# Kinesin-2 transports Orco into the olfactory cilium of *Drosophila melanogaster* at specific developmental stages

Swadhin Chandra Jana[1,2☯¤a]*, Priya Dutta[1☯], Akanksha Jain[1☯¤b], Anjusha Singh[1¤c], Lavanya Adusumilli[1¤d], Mukul Girotra[1¤e], Diksha Kumari[1], Seema Shirolikar[1], Krishanu Ray[1]*

1 Department of Biological Sciences, Tata Institute of Fundamental Research, Mumbai, India, 2 Instituto Gulbenkian de Ciência (IGC), Oeiras, Portugal

☯ These authors contributed equally to this work.
¤a Current address: National Centre for Biological Sciences, TIFR, Bengaluru, India
¤b Current address: Department of Biosystems Science and Engineering, ETH-Zurich, Basel, Switzerland
¤c Current address: Prescient HEALTHCARE Group, Gurgaon, India
¤d Current address: Bristol Myers Squibb, Redwood City, California, United States of America
¤e Current address: Ludwig Institute for Cancer Research, University of Lausanne, Switzerland
* swadhin1cal@gmail.com (SCJ); krishanu@tifr.res.in, krishanu64@gmail.com (KR)

**Data Availability Statement:** Raw datasets of the plots and histograms are uploaded as Supplemental Dataset.

## Abstract

The cilium, the sensing centre for the cell, displays an extensive repertoire of receptors for various cell signalling processes. The dynamic nature of ciliary signalling indicates that the ciliary entry of receptors and associated proteins must be regulated and conditional. To understand this process, we studied the ciliary localisation of the odour-receptor coreceptor (Orco), a seven-pass transmembrane protein essential for insect olfaction. Little is known about when and how Orco gets into the cilia. Here, using *Drosophila melanogaster*, we show that the bulk of Orco selectively enters the cilia on adult olfactory sensory neurons in two discrete, one-hour intervals after eclosion. A conditional loss of heterotrimeric kinesin-2 during this period reduces the electrophysiological response to odours and affects olfactory behaviour. We further show that Orco binds to the C-terminal tail fragments of the heterotrimeric kinesin-2 motor, which is required to transfer Orco from the ciliary base to the outer segment and maintain within an approximately four-micron stretch at the distal portion of the ciliary outer-segment. The Orco transport was not affected by the loss of critical intraflagellar transport components, IFT172/Oseg2 and IFT88/NompB, respectively, during the adult stage. These results highlight a novel developmental regulation of seven-pass transmembrane receptor transport into the cilia and indicate that ciliary signalling is both developmentally and temporally regulated.

## Author summary

Cilia are antenna-like organelles of eukaryotic cells, which mediate cellular signalling. Dysregulation of cilia assembly and the functionalisation process leads to numerous

**Funding:** The study was funded by the Department of Atomic Energy, Government of India, TIFR DAE grant nos. 12-R&D-TFR-5.10-100 and RIT4003 (Biol, Chem, Phys & Material Sc.) to K. R. The funders had no role in study design, data collection and analysis, decision to publish, or preparation of the manuscript.

**Competing interests:** The authors have declared that no competing interests exist.

human diseases, including ciliopathies and cancer. In sensory cilia, specific sub-compartments are established to support multiple signal transduction processes, such as photoreception, olfaction, and mechanosensation. In the last two decades, several reports described how a cilium is built and functions. However, a detailed understanding of the timing and modulation of receptor cargo transport to the cilium remains unclear. Here, we show that Orco, a signalling coreceptor for a family of insect olfactory receptors, specifically enters the cilia expressed on olfactory sensory neurons, and the process is restricted to a narrow developmental window in the fruit fly, *Drosophila*. The coreceptor is kept out of the cilia until it reaches its structural maturity shortly before *Drosophila* completes its metamorphosis and emerges from the pupal casing. This time-gated Orco entry would enable the adult fly to smell the environment as it emerged from the pupal case. We also find that the heterotrimeric kinesin-2, a core evolutionarily conserved anterograde motor essential for assembling the ciliary structure, is critical for functionalising the olfactory cilia. It interacts with Orco and transports the receptor into the cilium in an IFT-independent fashion. In addition, the motor appeared to deliver and maintain Orco at its designated slot in the ciliary outer-segment. Altogether, our results describe how activation of the olfactory cilia in a multicellular organism is tightly regulated. This system also provides an excellent tool and platform to understand the complex regulation of signalling components in cilia functionalisation, maintenance, and plasticity.

## Introduction

Cilium, a microtubule-based eukaryotic organelle, is the cellular antenna that discerns the environment and functions as a signalling hub [1]. The localisation of transmembrane proteins such as the channel proteins [2,3], the G-protein-coupled receptors (GPCR) [4], growth factor receptors [5] and other membrane components into the cilia defines the signalling properties. These proteins move into the ciliary compartment in response to environmental stimuli and cellular signalling [6,7,8]. One of the key impediments to this movement is a diffusion barrier at the cilium base that isolates both the ciliary membrane and cytoplasm from the plasma membrane and cellular cytoplasm, respectively [9,10]. Furthermore, the ciliary compartment is subdivided into multiple dynamic compartments with distinct lipid and protein compositions essential for cell signalling through the cilia [8,11–13]. The receptors are required to reach these compartments for activation of downstream signalling. Together this leaves us with the question—how the signalling receptors overcome these impediments and actively localise into these dynamic ciliary compartments maintaining the cilia function?

The issue is extensively investigated in the transport and localisation of the seven-pass transmembrane receptor Smoothened (Smo) in the cilia in tissue-cultured cells. Smo translocation to cilia dissociates the transcription factor Glioma-associated oncogene (Gli) from its repressor Suppressor of Fused (SuFu) in the cilium, allowing it to move into the nucleus. The kinesin-2 family motor, Kif3A, is implicated in the Hedgehog (Hh)-dependent Smo translocation into the primary cilium along with the β-arrestin [14]. The Gli-SuFu complex is localised at the ciliary tip, and Smo must reach this sub-compartment to disengage Gli. A separate study further suggested that the kinesin-family motor Costal2/Kif7, which is immotile and does not bind to either Smo or Gli, establishes the signalling-competent compartment at the ciliary tip [15]. Smo appeared to diffuse freely along the ciliary membrane after crossing the transition zone, and the process is altered by changes in the cholesterol levels downstream of a GPCR signalling [16]. However, it is still unclear how Smo or Gli-SuFu reaches the tip compartment.

Kinesin-2 family motors are implicated in several other transmembrane receptors localisation in the cilia. Kif17, a homodimeric kinesin-2, localises the Dopamine receptor 1 to the cilia in mouse Inner Medullary Collecting Duct 3 (IMCD3) cells [17] and the Cyclic-nucleotide-gated channel-1A (CNG1A) in Madin-Darby Canine Kidney (MDCK) cells [3], respectively. Also, interactions with a kinesin-2 motor were indicated to alter the channel functions of Polycistin-2 and Fibrocystin [18]. In all these cases, the site of kinesin-2 motor function, *i.e.*, whether it helps in crossing the transition zone or afterwards, is unclear. Both the IFT and the kinesin-2 family motors are essential for the assembly and maintenance of the ciliary cytoskeleton [19–21]. Kinesin-2 also drives the constitutive anterograde intraflagellar transport (IFT), moving an extensive repertoire of ciliary proteins into and inside a cilium and flagellum, which is intrinsic to the maintenance of these appendages [9]. Because of the dual roles of kinesin-2, it has remained a challenge to distinguish its functions between IFT and independent transport in the cilia assembly, compartmentalisation and function *in vivo*. Therefore, an understanding of signal regulation requires knowledge of the molecular mechanism underlying the ciliary entry and organisation of transmembrane proteins.

To address these issues, we investigated the transport of a transmembrane protein, the odour-receptor coreceptor (ORCO), into the olfactory sensory cilia in the adult *Drosophila* antenna. The third antennal segment in *Drosophila* contains multiple classes of sensilla. Each group of these sensilla are innervated with a specific set of ciliated olfactory sensory neurons (OSNs) expressing unique odorant receptors (ORs) [22]. More than 60 different ORs express in these OSNs [23]. Each ORx heterodimerises with Orco forming a functional odorant receptor complex essential to localise Orco/ORx to the olfactory cilia [24]. In this study, we describe Orco transport dynamics during cilia development and functional maturation. We show that independent of its role in cilia development, kinesin-2 transports Orco into the cilia selectively in two short episodes after the adult flies eclose from pupae. To demonstrate a critical role of heterotrimeric kinesin-2 in Orco/ORx localisation into the outer-segment (OS) of an olfactory cilium, we isolated a unique set of kinesin-2 mutants that selectively affect olfactory functions in adult *Drosophila* and established conditional manipulation of kinesin-2 function in the adult olfactory neurons. Also, we used affinity chromatography to demonstrate that Orco and OR47b could associate with the C-terminal tail domains of the kinesin-2 motor subunits. Together, the results indicate that an exclusive transport paradigm involving heterotrimeric kinesin-2 translocates the receptor from the dendritic knob and maintains them in an approximately 4 μm zone at the distal part of the ciliary outer-segment, which is distinct from the tip compartment marked by the kinesin-2 enrichment. The Orco localisation in the cilia may designate the odour signalling compartment.

## Results

### Orco selectively localises to the olfactory cilia

*Drosophila* antenna contains three morphologically distinct typed of cuticular projections, called *sensilla*, each innervated with the cilia of 2–4 OSNs [22,25]. Bipartite cilia extend from the distal ends of dendrites of the OSNs and grow into the hollow cuticular shafts of the *sensilla* (Fig 1A) [20]. Orco expresses in all the OSNs innervating *sensilla basiconica* and *sensilla trichodea* and localises in the cilia (Fig 1A). Association of Or22a and Or43a, respectively, with Orco, is essential for the Orco/ORx entry into the cilia [24]. Accordingly, ectopic overexpression of *GFP*:*Orco*, *GFP*:*Or47b* and *GFP*:*Or43a* (ORxs) in cholinergic neurons enriched the GFP-tagged proteins in both the basiconic and trichoid cilia (Fig 1Ad-f). Contrary to the expectation, the ectopically expressed Or43a and Orco were excluded from the mechanosensory cilia on chordotonal neurons, innervating the Johnston's organ in the second antennal

segment (Fig 1B). Together these suggested that Orco/ORx specifically enters the olfactory sensory cilia, and molecular systems unique to the OSNs facilitate the ciliary access. A similar phenomenon was observed for different G-protein coupled receptors in *C. elegans* [8].

The olfactory cilium extends from a basal body (BB) placed at the distal end of the OSN dendrite (Fig 1C). Each cilium inside the *s. basiconica* shaft, henceforth called basiconic cilia, has a bipartite organisation well-resolved by confocal microscopy. The base consists of a 9+0 arrangement of microtubule doublets (Fig 1D), which supports a short (~0.5 μm) connecting cilium (CC) between the ciliary inner-segment (IS) and the basal body (BB). The outer-segment (OS) is extended from the IS and housed within the porous cuticle shaft. It consists of membranous branches supported by singlet microtubules (Fig 1Dd-e). The OS contains two distinct structural domains. The proximal domain with fewer branches containing multiple microtubules each, and the distal domain where each branch has a singlet microtubule tightly invested with the ciliary membrane (Fig 1Dd-e). The ab1-type large *s. basiconica* are innervated with four OSNs, and one of these expresses OR42b [26]. We used these sensilla because of their characteristic features and high levels of Orco enrichment (Fig 1E and 1F) for all quantitative analyses in this study.

## The bulk of the Orco enters cilia in two brief episodes, shortly after eclosion

Olfactory cilia develop in stages, and the process completes by 90-hours after pupa formation (APF) [20]. Consistent with anti-Orco immunostaining analysis (S1A Fig), the GFP-tagged Orco, expressed from its endogenous promoter in the *orco^fTRG* stock [27], hereafter, referred to as Endo-Orco:GFP, marked a few basiconic cilia after 90 hours APF. As the development progressed, Endo-Orco:GFP appeared in an increasing number of basiconic cilia (S1B and S1C Fig). The onset of Orco expression, which begins from 80 hours APF [28], could determine the timing of its entry into the cilium. To test the conjecture, we ectopically expressed GFP:Orco, GFP:Or47b, and coexpressed GFP:Orco with GFP:Or43a using *chaGal4*, which marked the OSNs from an earlier stage of development. In all these cases, the GFP-tagged transgenic proteins were restricted to the cell body until 96-hours APF (S1D Fig). In comparison, the expression of soluble GFP and mCD8:GFP using the same driver marked the entire ciliary volume and membrane, respectively, throughout the pupal stages (S1D Fig). These results suggested that the Orco/ORx entry into the olfactory cilium is regulated independently of their gene expression profiles. A similar phenomenon was observed using *orcoGal4>GFP*: *Orco* background (S1E Fig).

The ciliary levels of Endo-Orco:GFP reached the steady-state within 6 hours after eclosion (AE; S1F–S1G Fig), indicating that the bulk of Orco/ORx enters the cilia within a short period immediately after the adult emerged from the pupal case. We further resolved the entry period to two distinct phases in the ab1-type sensilla (Figs 1G and S2A). The same pattern was observed in the cilium expressed on Or42b OSN (Fig 1G). The episodic Orco entry pattern was exclusive. The bulk membrane content, monitored using the heterologous bacterial transmembrane protein, ReaChR:YFP, a chimeric channelrhodopsin (Fig 1G), and the ciliary volume remained unaltered during the Orco entry period (S2B and S2C Fig). Altogether, these observations suggested that Orco/ORx independently enters the cilium after it is fully developed and the process is developmentally gated. Hence, it raised a new question–how is the Orco entry regulated?

## Kinesin-2 is involved in maintaining odour reception independent of the cilia assembly

The cilia structure and function are intimately linked [7]. Genetic studies suggested that the heterotrimeric kinesin-2 is essential for cilia assembly and maintenance [29] and the ciliary

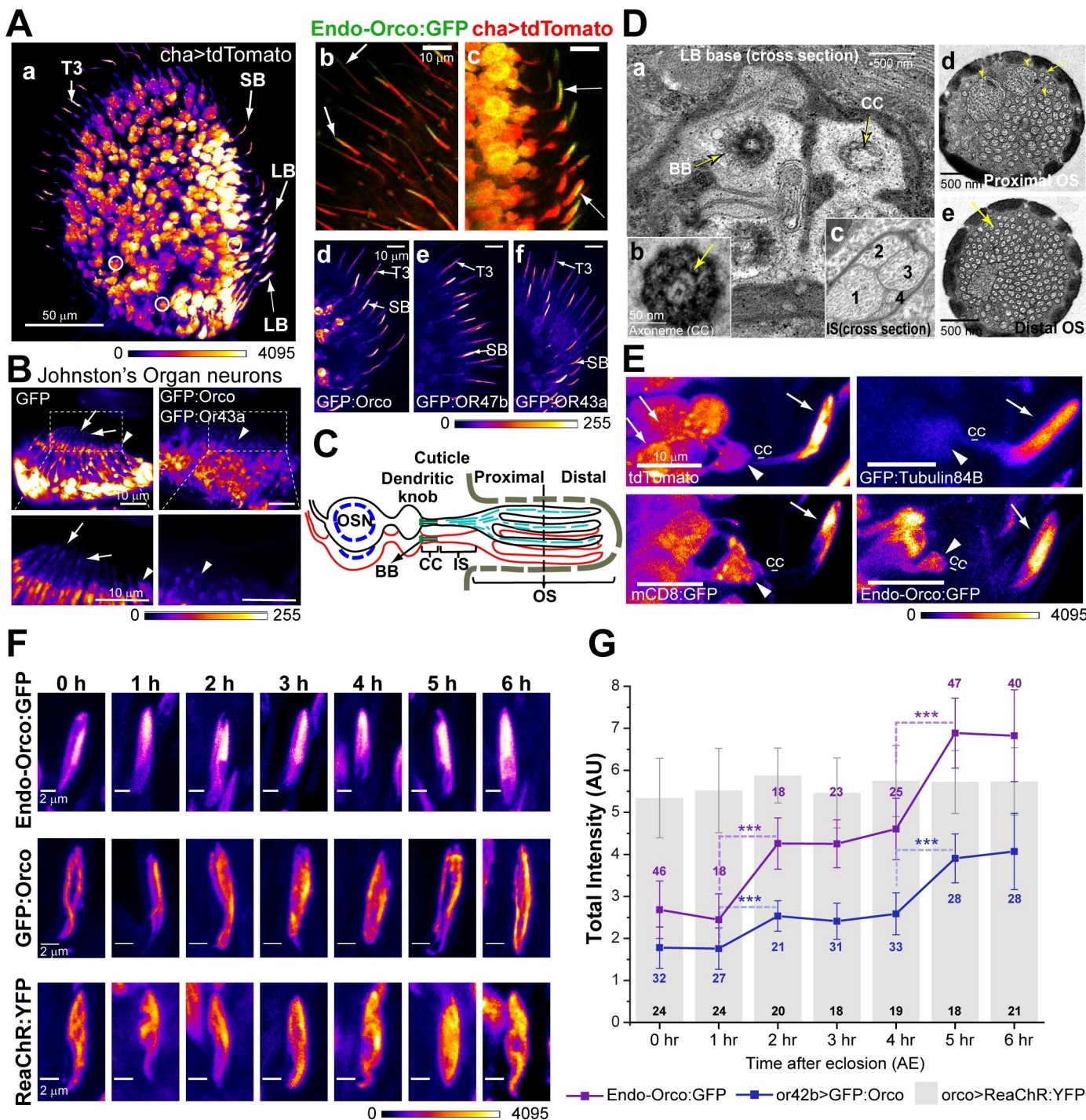

**Fig 1. The developmental profile of Orco localisation in the cilia.** (A) Organization of olfactory sensory neurons (OSNs) in the adult third antennal segment marked by *chaGal4>UAS-tdTomato* (a), and enlarged views of the *s. trichodea* (b) and *s. basiconica* (c) regions expressing both Endo-Orco:GFP (*orco^fTRG*) and *chaGal4>UAS-tdTomato*. A few OSN cell bodies are circled white, and the labelled arrows indicate cilia inside small and large *s. basiconica* (SB and LB type) and *s. trichodea* (T3 type) shafts. Ectopic expressions of GFP:Orco (d), GFP:Or47b (e), and GFP:Or43a (f), respectively, by the *chaGal4*, marked the cilia (arrows) inside all SB and T3-type sensilla. (B) Chordotonal neurons of the Johnston's organ expressing soluble GFP and GFP:Orco along with GFP:Or43a, respectively, by *chaGal4*. Arrowheads mark the dendritic knobs, and arrows mark the cilia region. (C) Schematic illustrates cilia organisation on OSNs innervating *s. basiconica*. (D) TEM images of transverse sections through the base of an LB-type sensillum show the basal bodies (BB) and the connecting cilium (CC) of four innervating cilia (a). Each connecting cilium (CC) carries nine doublet microtubule (9+0) bearing axoneme (arrow, b). Sections of the inner segment (IS, c) and outer segment (OS, d, e) shows the arrangement of singlet microtubules within a membrane sleeve (yellow arrow). (E) tdTomato, GFP:Tubulin84B, mCD8:GFP and Endo-Orco: GFP localisations in the cilia inside ab1-type *s. basiconica* (arrows- ciliary OS, and arrowheads- ciliary base). (F) Endo-Orco:GFP and the recombinant bacterial

channelrhodopsin (ReaChR:YFP) localisations in the ab1-type *s. basiconica* and that of ectopic GFP:Orco in the or42b cilium. ReaChR:YFP expressed using *ocroGal4* was used as a negative control. (G) Total fluorescence intensities (mean ± S.D.) of Endo-Orco:GFP and Orco:GFP enrichments in the cilia inside ab1-type sensilla at different ages after eclosion. The pairwise significance of difference was estimated using one-way ANOVA test, p-values (*p < 0.05, **p < 0.01, and ***p<0.001) are shown on the plots. Images are shown in a false colour intensity heat map (FIRE, ImageJ). Scale bars indicate 50 μm (A-a), 10 μm (A- b-f, B, and E), and 2 μm (F), respectively, and the number of sensilla (n) used for the quantifications is indicated on the plots with appropriate colour coding.

enrichment of several transmembrane receptors [3,14,18,30]. For example, total loss-of-function (null) mutation in the *Klp64D* gene, which codes for the kinesin-2α/Kif3A motor subunit in *Drosophila*, disrupts the development of olfactory cilia as well as reduces the odour-evoked potentials from the antenna [20]. Therefore, to assess whether the motor is separately engaged in the Orco/ORx localisation and transport, we first utilised a forward genetic strategy to disrupt the olfaction-specific function of kinesin-2. We isolated viable *Klp64D* alleles with selective olfaction defects in the adult stage (Fig 2A). An initial screen for jump response defects in adult flies isolated five homozygous, viable *Klp64D* mutant alleles (*kj353*, *kj429*, *kj925*, *kj1070*, and *kj1072*) with substantial olfactory defects (Fig 2B and 2C). The chemotactic behaviour of the mutant larvae towards ethyl-acetate and Butanol (S1 Table), the locomotion of the homozygous mutant adults, and the ultrastructure of the cilia on chordotonal neurons in Johnston's organ were unaffected in the homozygous *Klp64D^{kj}* backgrounds (S3A and S3B Fig). These observations confirmed that the *Klp64D^{kj}* mutations selectively disrupt adult olfaction, fulfilling the central objective of the screen.

The olfactory sensory responses and the odour-induced neuronal activity are measured using the Jump assay and the Electroantennogram (EAG). The aberrant odour-evoked jump responses (Fig 2B) and EAG defects (Fig 2C) displayed by the homozygous mutant flies were recessive in nature. We mapped them to the *Klp64D* locus using genetic non-complementation and transgenic rescue analyses (S2 Table). The olfactory defect of the mutants could be rescued by the ectopic expression of *Klp64D* transgene and not by the *Klp68D* (kinesin-2β/Kif3B orthologue) expression (S3C–S3E Fig). Together, these established that mutations in all the *kj* alleles belong to the *Klp64D* locus. Since the jump response and EAG characteristics of these alleles appeared very similar, we decided to continue further phenotypic characterisations using the kj353 and kj925 alleles. Both these alleles have good homozygous viabilities.

The mutation reduced the EAG responses to wide-ranging odours in the homozygous *Kp64D^{kj}* mutants (Fig 2C and 2D). Of these, ethyl acetate (EA), nButanol (nBut), and benzaldehyde (Benz) are detected by OSNs innervating *s. basiconica*, and propionic acid (PA) by OSNs innervating *s. coeloconica* [26,31]. These results suggested that the KLP64D function is necessary to develop odour reception in different types of olfactory cilia. A constant downshift of the response voltage at different concentrations of a variety of odorants further suggested that the loss of kinesin-2 function could suppress odour reception at different types of OSNs (Fig 2D).

KLP64D is essential for the heterotrimeric kinesin-2 assembly [32], and the level of KLP68D:YFP is significantly reduced in the cilia in the *Klp64D* total-loss-of-function allele [20]. Therefore, to understand the nature of the kinesin-2 disruption in *Klp64D^{kj}* alleles, we estimated the ciliary levels of KLP68D:YFP during development in wild-type and homozygous *Klp64D^{kj353}* and *Klp64D^{kj925}* backgrounds, respectively. The KLP68D:YFP level in the mutants was comparable to that of the wild-type at 96 hours APF, but it was significantly reduced in the freshly eclosed (0–3 hour AE) homozygous mutant adults (Fig 2E and 2F). The cuticle auto-fluorescence mostly contributed to the values estimated in the mutants in the adult stage (Fig 2F). This observation suggested that the mutations in homozygous *Klp64D^{kj}* alleles selectively reduce kinesin-2 entry into the olfactory cilia only in the adult stage.

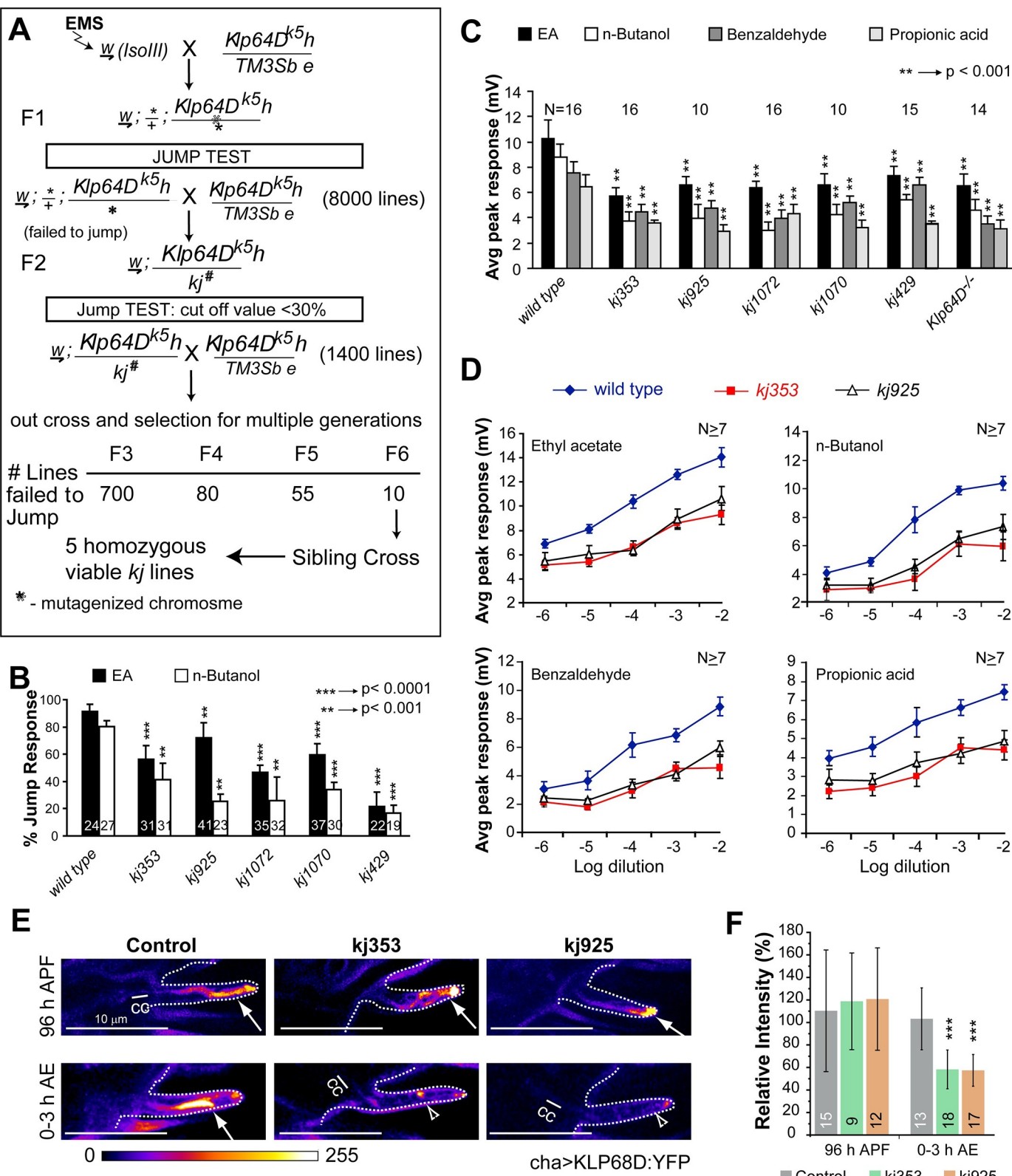

**Fig 2. Isolation of homozygous viable *Klp64D* mutant alleles with specific odour response defects in adults.** (A) Flow diagram summarises the ethyl-methane-sulfonate (EMS) mutagenesis screen used for isolating the viable alleles of Klp64D with jump response defect. $Klp64D^{k5}$ is a homozygous lethal allele. The freshly mutagenised genome was screened for jump response defect mutants in trans-heterozygote combinations over the $Klp64D^{k5}$ for several generations. Fx (x = 1 to 6)

indicates the number of generations of outcross with $Klp64D^{k5}$. (B) Jump response indices of wild type (Canton S) and homozygous *kj* isolates. (C) Electroantennogram (EAG) responses to different odours measured from the homozygous *kj* mutants (see S2 Table and S3 Fig for detail genetic characterisation of the *kj* alleles by EAG). The pairwise significance of difference was estimated using two-tailed Student's T-test, p-values (*p < 0.05, **p < 0.01, and ***p<0.001) are shown on the plots. (D) EAG response analyses at different dilutions of four different odorants across the spectrum show that mutations in the homozygous *kj* flies lower the odour sensitivity (N ≥10 or as depicted on each set of bars). (E) KLP68D:YFP localisation in the cilia (expressed by chaGal4) in control, and homozygous *kj353* and *kj925* backgrounds, at 96 hours APF and 0–3 hours AE, respectively. Connecting cilia (CC), and the outer segments (OS, arrows) are shown in each panel (Scale, 10 μm). Images are shown using a false-colour intensity heat map (FIRE, ImageJ). (F) Relative intensity values of the KLP68D:YFP fluorescence in the cilia estimated against the control group at each stage. Number of sensillae (n) and the number of antennae (N) are indicated on the plots.

KIF17, the homodimeric kinesin-2, has been reported to facilitate the localisation of the cyclic nucleotide gated channel comprising CNGA2 and CNGB1b subunits in the mammalian olfactory cilia [3]. Therefore, we reasoned that the loss of kinesin-2 entry into the cilia could affect the Orco/ORx transport resulting in the loss of odour reception. However, Kif17 is redundant for the primary cilia assembly, whereas the heterotrimeric kinesin-2 also transports tubulin into the olfactory cilia [33] and powers the IFT [29]. Both are essential for maintaining the cytoskeletal integrity in the cilia. Hence, alternatively, disruption of the ciliary cytoskeleton due to the loss of kinesin-2 function could cause the EAG defect as reported earlier [20].

## Loss of kinesin-2 in the adult stage reduced the cilia Orco enrichment

To distinguish between these two possibilities, we compared the ciliary enrichment of GFP:Orco, GFP:Tubulin84B, and a non-specific membrane marker (mCD8:RFP), respectively, in wild type and homozygous $Klp64D^{kj}$ mutants. The GFP:Orco levels, independently expressed in the OSNs by *chaGal4* and *orcoGal4*, were significantly reduced in homozygous $Klp64D^{kj353}$ adults (Fig 3A and 3B), and the expression of KLP64D transgene in the $Klp64D^{kj353}$ mutant background rescued the defect (Fig 3C). The levels were unaffected at 96 hours APF. These observations suggested that the mutations in the *Klp64D* gene restricted the GFP:Orco enrichment in adult cilia. Also, we found that the total GFP:Tubulin84B level and the ciliary volume were unaltered, and there was a marginal increase in the cytoplasmic GFP and mCD8:RFP levels in the cilia in homozygous $Klp64D^{kj353}$ background (S4A Fig).

To understand the underlying cause of this Orco reduction, we studied the ciliary ultrastructure using TEM. The analysis of 1-day-old homozygous $Klp64D^{kj353}$ and $Klp64D^{k925}$ mutant adults revealed marginal defects in the ciliary ultrastructure (Fig 3D) with a small but significant reduction in the number of singlet microtubules only in the $Klp64D^{kj353}$ background (Fig 3F). Also, the number of membrane-bound branches bearing singlet microtubules were considerably reduced in the homozygous $Klp64D^{kj}$ backgrounds (Fig 3G). The defect was much less severe than those observed in the homozygous $Klp64D^{k1}$ background [20], which is recessive lethal and considered to be a null allele. The apparent difference in the microtubule count in the TEM sections and the total GFP:Tubulin84B levels assessed from the confocal images could be attributed to the difference in resolutions of these two techniques. Together, these results suggested that loss of kinesin-2 function in the adult olfactory cilia has a relatively stronger impact on Orco localisation than on the axoneme.

The ciliary tubulin has low turnover during the adult stage [33]. Consistent with this report, very little GFP:Tubulin84B localised in the cilia when expressed using *orcoGal4* as compared to the *chaGal4* and *or42bGal4*, respectively (S4B Fig). *OrcoGal4* expresses after 90 hours APF, much later than the onset of *chaGal4* (30–34 hours APF) and *or42bGal4* (60–70 hours APF). Hence, the results suggested that very little tubulin is transported into the cilia in the adult stage. Together, these observations explained the relatively milder impact on microtubule count in the homozygous $Klp64D^{kj}$ backgrounds. Therefore, we concluded that the loss of

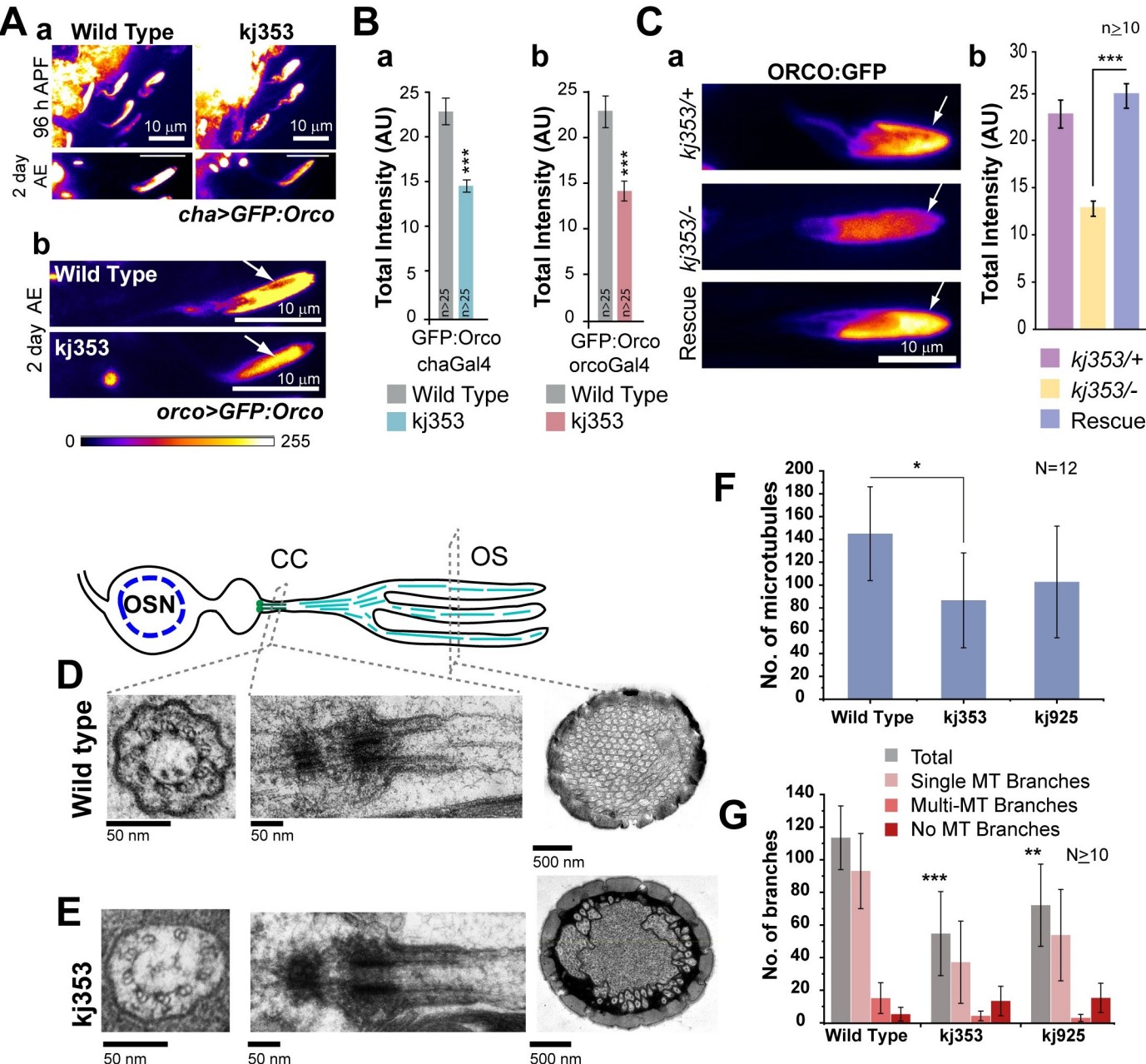

**Fig 3. Mutations in the *Klp64D* gene reduced Orco localisation in the adult olfactory cilia.** (A) GFP:Orco localisation in cilia of *s. basiconica* (arrow), in 96-h APF and two-day-old control and homozygous *Klp64D^{kj353}* mutant antennae. The GFP:Orco transgene was separately expressed using *chaGal4* (a) and *orcoGal4* (b) to test the effects of different levels of recombinant protein expression on its cilia localisation. (B) Total fluorescence intensity of GFP:Orco, driven by *chaGal4* (a) and *orcoGal4* (b) in the cilia of *s. basiconica* of two-days-old control and homozygous *Klp64D^{kj353}* mutant adults. (C) GFP:Orco localisation in the ciliary OS of control (*chaGal4>UAS-GFP:Orco/+; Klp64D^{kj353}/+*), hemizygous *kj353* (*chaGal4>UAS-GFP:Orco/+; kj353/Klp64D^{k1}*) and rescued (*chaGal4>UAS-GFP:Orco/ UAS-KLP64D-TevHis; Klp64D^{kj353}/Klp64D^{k1}*) adults (a). Total fluorescence intensity (mean ± S.D.) measured in the different genetic backgrounds (b). The pairwise significance of difference was estimated using two-tailed Student's T-test, p-values (*p < 0.05, **p < 0.01, and ***p<0.001) are shown on the plots. All Images are shown in a false colour intensity heat map (FIRE, ImageJ), and scale bars indicate 10 μm. (D-E) Ultrastructure of the basiconic cilia in the wild-type and homozygous *Klp64D^{kj}* antennae. Schematic illustrates the microtubule organisation in cilia inside *s. basiconica*. TEM images of sections through the basal bodies, connecting cilia (CC) and outer segments (OS) of cilia inside *s. basiconica* from one-day-old wild-type (D) and homozygous kj353 (E) antennae. F, G) Histograms describe the total number of singlet MTs (F) and the distribution of different types of branches (G) found in the OS in wild type and the mutants. The pairwise significance of difference was estimated using one-way ANOVA test, p-values (*p < 0.05, **p < 0.01, and ***p<0.001) are shown on the plots. Error bars represent as ± S. D. The n values indicate number of sensilla in panels B and C, and the N values in panel F indicate number of TEM sections obtained from at least 3 different heads.

kinesin-2 function in the adult OSN would not significantly affect the ciliary cytoskeleton in the near term. The loss of membrane branches was, however, intriguing.

## Orco entry and transport into the ciliary outer-segment requires kinesin-2

To assess the function of kinesin-2 in the bi-phasic increment of Orco during 0–6 hours AE, we reduced the kinesin-2 levels just before eclosion. The KLP64D and KLP68D dsRNA were expressed using *orcoGal4* in the *orco^fTRG* background, to suppress the target genes in the late pupa (after 90 hours APF) and adult stages. We found a significant reduction of Endo-Orco: GFP level in the cilia at 0 hour AE (~105 hours APF) in both the Klp64D and Klp68D RNAi backgrounds. Although the level appreciated to some extent in the subsequent stages, it remained significantly lower as compared to the control until 12 hours AE (Fig 4B, 4C and 4E). A similar profile was obtained in the *Klp64D^kj353* homozygous background (Figs 4D, 4E and S5A). The reduction of Endo-Orco:GFP in the cilia in the kinesin-2 RNAi and mutant backgrounds was associated with its episodic accumulation at the dendritic-knob region at 2, 4, and 6 hours AE (Fig 4B–4D), suggesting that the reduction of kinesin-2 levels affect Orco movement from the dendritic base into the cilium. In comparison, the Jupiter:GFP levels, an indicator of the ciliary microtubule content, remained unaffected in these backgrounds (S5C Fig). Together, these observations implicated kinesin-2 in the Orco movement through the transition zone.

After the Orco transfer to the ciliary membrane, it could potentially diffuse throughout the cilium. In such a scenario, one would expect a uniform distribution of the protein along the cilium. However, like Smo [14] and the salt-sensing receptor-type guanylate cyclase (GCY-22) [34], Orco is distinctly enriched in the ciliary OS. Therefore, to explore a kinesin-2 role in the Orco movement inside the cilia, we used the Endo-Orco:GFP fluorescence recovery after photobleaching (FRAP) at the distal half of the ciliary OS during 4–5 hours AE, which coincided with the second Orco enrichment episode. In a majority of the wild-type cilia (n = 8), the fast recovery stage completed within 20 minutes (S6B Fig) and in some cilia, the FRAP profiles had multiple episodic increments (n = 3), each lasting for 10–20 minutes (S6B Fig). The episodic nature of the recovery indicated that the Orco movement occurs inside the cilia through active transport. The recovery pattern appeared divergent because it was difficult to synchronise the photobleaching time with the ageing process.

In comparison, the recovery profiles in the kinesin-2 RNAi backgrounds were visibly slower, with no clearly discernible episodic profiles (S6B Fig). Although some episodic increments were observed in homozygous *Klp64D^kj353* background (n = 2), the extent of recovery did not exceed 30%, which is significantly below the minimum recovery level observed in the wild-type control (Figs 4E and S6B). Together, these data suggested that Orco movement inside the cilium would require kinesin-2. Furthermore, the relatively slow rate of initial recovery in the kinesin-2 mutants might indicate that Orco diffusion in the cilium would be highly restricted.

## Orco localises in the cilia independent of IFT172/Oseg2 and IFT88/NompB

Several receptors, for example, GCY-22, move inside the cilium through IFT [34], and Oseg2/ IFT172 is particularly implicated in the transmembrane receptor transport into the cilia [35]. IFT movements are most prominent in the middle segment of the amphid and AWC cilia {Scheidel, 2018 #115}. The episodic Orco recovery during 0-6-h AE suggested a similar mode of movement. To explore whether Orco indeed moves in the cilia in an IFT-like manner, we compared the Endo-Orco:GFP and GFP:Orco localisation patterns with that of the IFT-B component Oseg2:GFP and KLP64D:GFP at the ciliary inner-segment (IS) in this period. We found prominent Oseg2:GFP puncta representing the IFT trains in the ciliary IS (arrowheads,

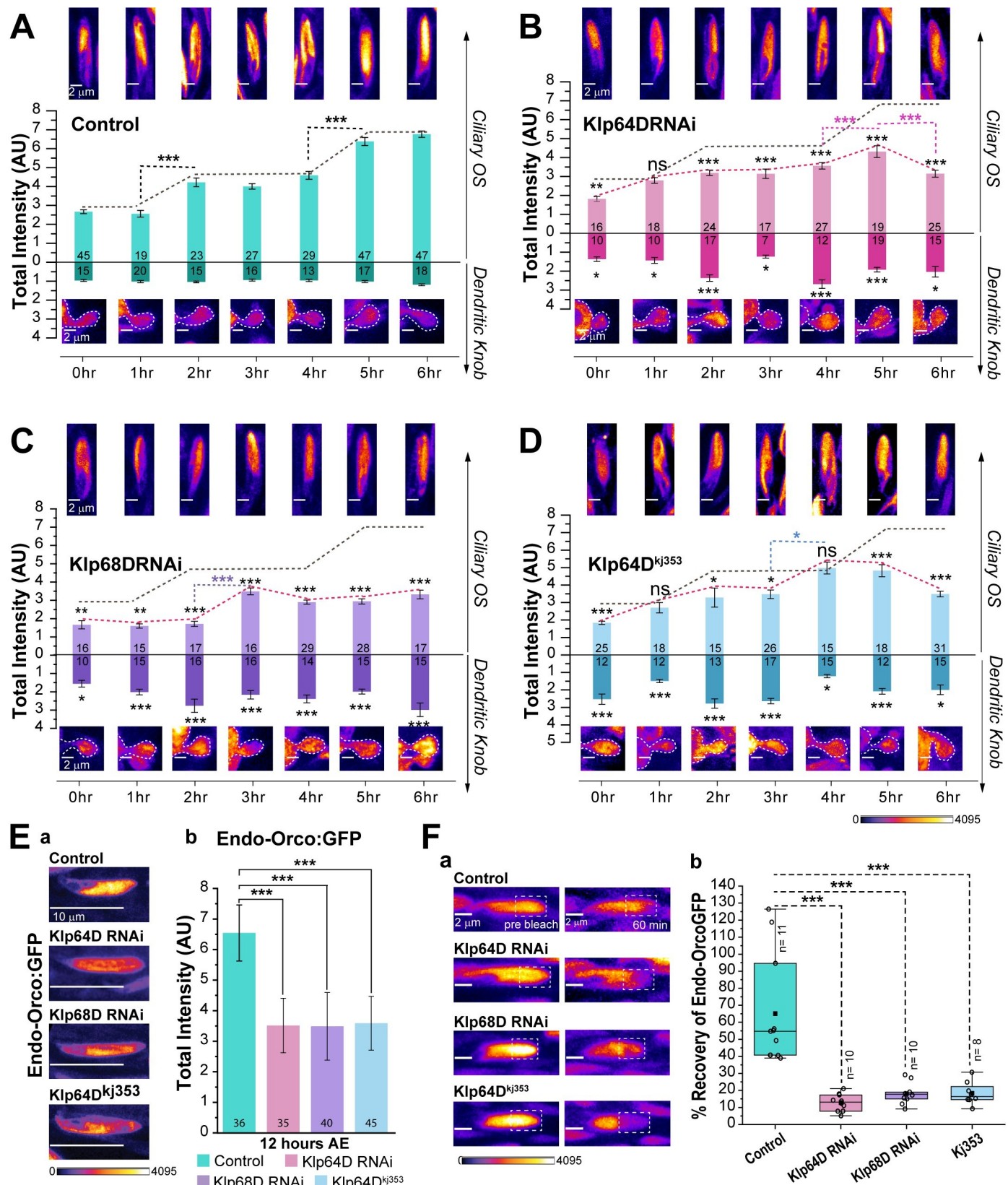

**Fig 4. Kinesin-2 is necessary for Orco entry into the olfactory cilia.** (A-D) Total fluorescence intensity (mean ± S.E.M) of Endo-Orco:GFP in the ciliary OS and the dendritic knob region of OSNs in Control (A), Klp64D RNAi (B), Klp68D RNAi (C) and homozygous *Klp64D^kj353* mutant (D) backgrounds, respectively. The pairwise significance of difference as compared to the wild type control and between different stages was estimated using one-way ANOVA test, and the p-values

($^*$p < 0.05, $^{**}$p < 0.01, and $^{***}$p<0.001) are shown on the plots with appropriate colour codes. Inset shows representative images of the cilia at corresponding stages. (E) Images depict Endo-Orco:GFP localisation in the ab1-type *s. basiconica* from Control (*orcobGal4, Endo-Orco:GFP/+; UAS-Dicer/+*), Klp64D RNAi (*orcoGal4, Endo-Orco:GFP/UAS-Klp64D RNAi; UAS-Dicer/+*), Klp68D RNAi (*orcoGal4, Endo-Orco:GFP/UAS-Klp68D RNAi; UAS-Dicer/+*) and homozygous *Klp64D$^{kj353}$* mutant (*Endo-Orco:GFP; Klp64D$^{kj353}$*) backgrounds at 12 hours AE (a). The histograms depict total fluorescence intensity (mean ± S.D.) of Endo-Orco: GFP in control, the kinesin-2 knock downs, and homozygous *Klp64D$^{kj353}$* mutant backgrounds at 12 hours AE (b). The pairwise significance of difference was estimated using the one-way ANOVA test, and p-values ($^*$p < 0.05, $^{**}$p < 0.01, and $^{***}$p<0.001) are indicated on the plots. (F) The recovery of Endo-Orco:GFP fluorescence in the bleached region of the cilia at 60 minutes after the photobleaching as compared to the unbleached region in the Control, Klp64D RNAi, Klp68D RNAi and homozygous *Klp64D$^{kj353}$* mutant backgrounds (also see S6 Fig). The cilia were photobleached during 4–5 hours AE. Examples of prebelach and postbleach images are shown in panel a, and the corresponding quantification is presented in panel b. The pairwise significance of difference as compared to the wild type control was estimated using Mann-Whitney U-test, and the p-values ($^*$p < 0.05, $^{**}$p < 0.01, and $^{***}$p<0.001) are indicated on the box plots. All images are shown in the false colour intensity heat map (FIRE, ImageJ). Scale for images—2 μm (A-D), 10 μm (E-a), and 2 μm (F-a). Number of sensillae (n) quantified is indicated on the graphs.

Fig 5A, N = 4, n = 7), whereas the KLP64D:GFP distribution was relatively more uniform (arrows, Fig 5A, N = 5, n = 7). No such GFP:Orco puncta were detected in the IS region in the overexpressed GFP:Orco (N = 6, n = 9), as well as the EndoOrco:GFP in wild-type (N = 5, n = 8) and homozygous *Klp64D$^{kj353}$* mutant (N = 4, n = 9) backgrounds (Fig 5A). Furthermore, the adult OSN-specific RNAi of NompB/IFT88 and Oseg2/IFT172 did not affect the Endo-Orco:GFP localisation in the cilia (Fig 5B). The RNAi, however, eliminated both the Oseg2:GFP (Fig 5C) and an IFT-A complex component Oseg4/WDR35:GFP (Fig 5D) from the adult olfactory cilia. Together, these results suggested that the Orco entry through the transition zone and movement inside the cilium is unlikely to involve IFT.

## Orco associates with the tail domains of kinesin-2 motor subunits

To supplement the genetic evidence, we then used copurification assay to test whether the heterotrimeric kinesin-2 motor (Fig 6A) could directly engage with Orco/ORx. The N-terminal domains of kinesin-2 motor subunits bind to the microtubule and generate the force [29]. The coiled-coil stalk domains in the middle are involved in heterodimerisation [32] and bind to the accessory subunit, kinesin associated protein (KAP) [36], and an IFT complex subunit [37]. Both the KLP68D (kinesin-2β) and DmKAP (Fig 6B) were pulled down from the head extracts of *chaGal4>UAS-GFP:Orco* adults using agarose beads conjugated with GFP-nanobody (chromotek, USA), suggesting that Orco interacts with the heterotrimeric kinesin-2. Further, a relatively more prominent level of GFP:Orco (Fig 6C) and OR47b, respectively, were pulled-down using the GST-KLP68D-Tail fragment (Fig 6D), which identified the potential Orco/ORx binding region at the tail domain of KLP68D (2β) motor subunit. Together, these results suggested that Orco/ORx associates with the heterotrimeric kinesin-2 motor, which subsequently transports it into the cilium.

## Orco enriches in a characteristic distal domain at the ciliary OS

Next, we examined how Orco localises inside the cilium after crossing the transition zone. The time-resolved plot profile analysis revealed that Orco is mostly enriched in the distal OS, in an approximately 4 μm band with a stereotypic peak at the distal OS (Fig 7A and 7B). This portion of the cilia, housed within the porous shaft of basiconica, consists of predominantly singlet microtubule-bearing branches [20,25]. Similar Orco localisation, distinct from that of the tubulin, was observed in the Or42b cilium (Fig 7C and 7D). The distribution of Orco was maintained throughout 0–6 hours AE, and it was distinct from that of the membrane marked by Bodipy-FL-C$_{12}$ (Fig 7E and 7F). The cytoskeleton marked by *chaGal4>UAS-GFP:tubulin84B* and endogenous Jupiter:GFP (MAP) (S7B Fig), as well as the cytoplasm labelled by GFP (S7C Fig), also elicited a relatively uniform distribution profile throughout the OS (Figs 7F and S7). Thus, the enrichment domain appeared exclusive for Orco. Interestingly, the mCD8:GFP,

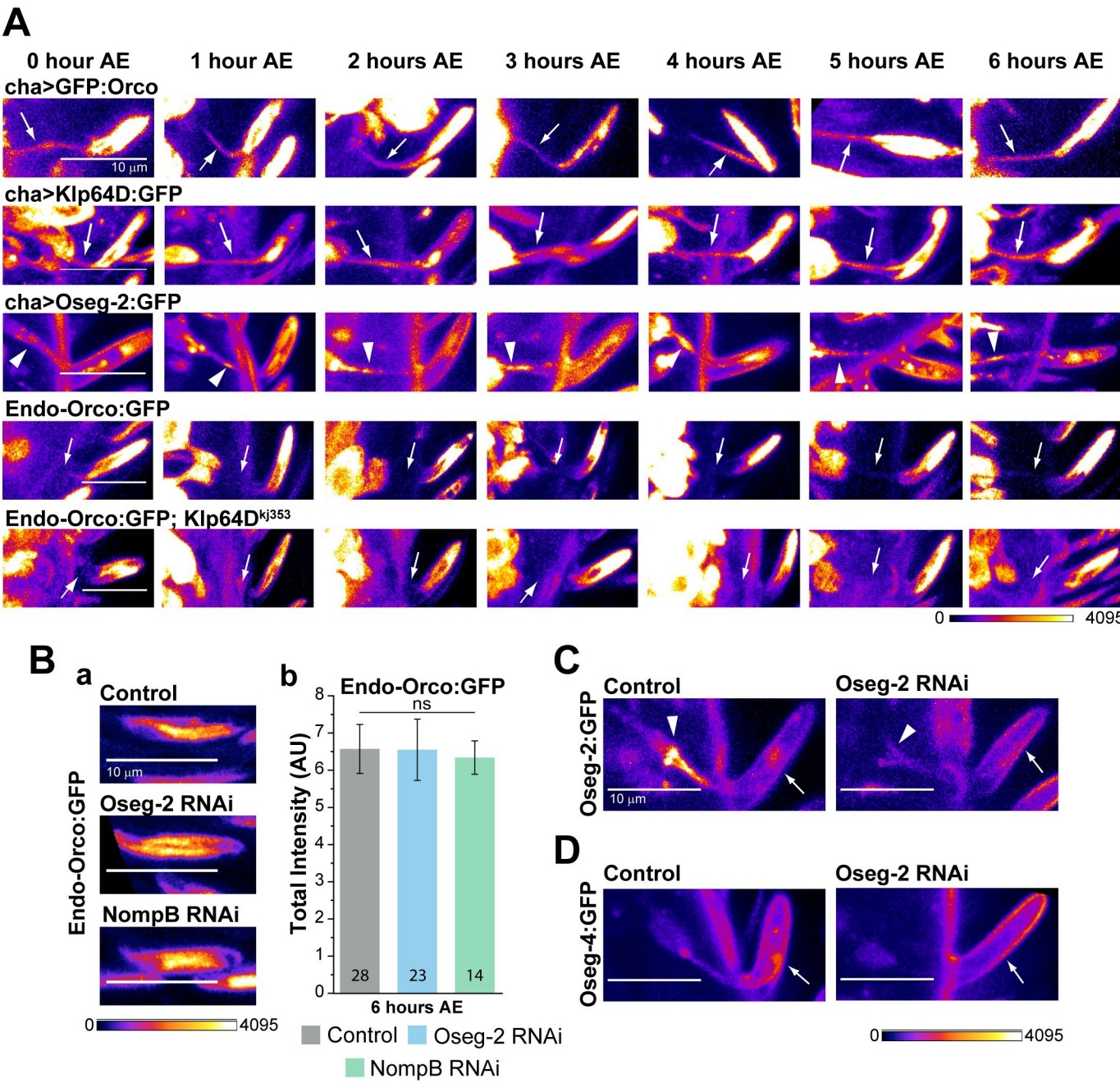

**Fig 5. IFT172/Oseg2 is not required for Orco enrichment in the olfactory cilia.** (A) The patterns of GFP:Orco, Oseg2:GFP, KLP64:GFP, and Endo-Orco:GFP localizations along the olfactory cilia during 0–6 hour AE in the wild-type (*cha>*) and *Klp64D^{kj353}* mutant backgrounds. Fine arrows indicate relatively uniform localisation of KLP64D:GFP, Endo-Orco:GFP and GFP:Orco and the arrowheads indicate clearly discernible Oseg2:GFP puctae in the inner segment. (B) Endo-Orco:GFP localisation in ab1-type *s. basiconica* from the control (*orcoGal4, Endo-Orco:GFP/+; UAS-Dicer/+*), Oseg2 RNAi (*orcoGal4, Endo-Orco:GFP/UAS-Oseg2 RNAi; UAS-Dicer/+*) and NompB RNAi (*orcoGal4, Endo-Orco:GFP/UAS-NompB RNAi; UAS-Dicer/+*) backgrounds, respectively, at 6 hours AE (a). Histograms indicate Endo-Orco:GFP fluorescence intensity (mean ± S.D.) in the cilia in the control, Oseg2 and NompB RNAi backgrounds at 6 hours AE (b). The pairwise significance of difference was estimated using one-way ANOVA, and the p-values (*p < 0.05, **p < 0.01, and ***p<0.001) are shown on the plots. (C, D) Oseg2:GFP and Oseg4:GFP localisations at the basal body (arrowhead) and cilia (arrow) in the control and Oseg-2 RNAi (driven by *orcoGal4*) backgrounds. Images are shown in a false colour intensity heat map (FIRE, ImageJ), arrowheads indicate the dendritic knob, arrows indicate the OS and scale bars indicate 10 μm. Number of sensillae (n) quantified is indicated on the graphs.

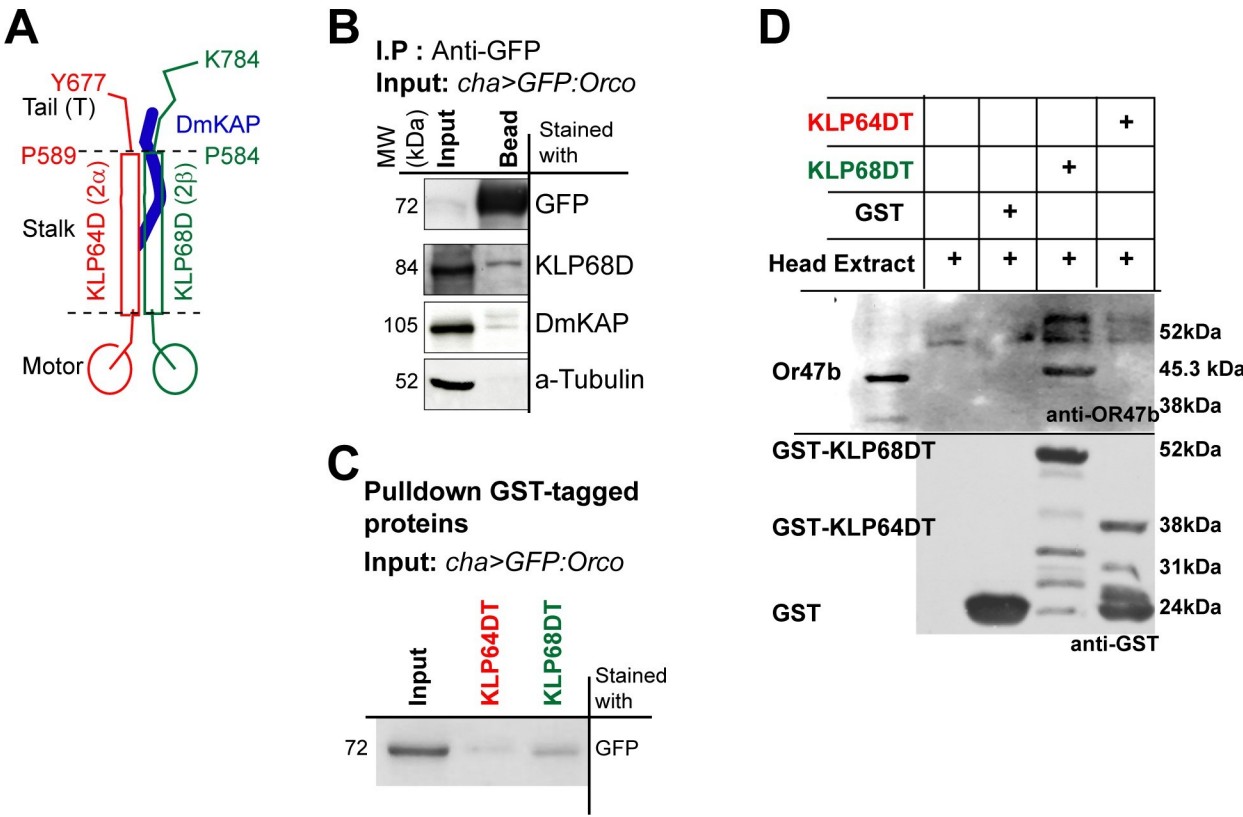

**Fig 6. Orco associates with the C-terminal 'tail' domain of the kinesin-2 motor subunits.** (A) Schematic illustrates the organisation of *Drosophila* kinesin-2 and the tail domains of the motor subunits. (B) Co-immunoprecipitation of the *Drosophila* kinesin-2 subunits from the head extracts of flies expressing GFP:Orco (driven by *chaGal4*) using agarose beads conjugated with GFP-nanobody. (C, D) Affinity copurification of GFP:Orco (C), and OR47b (D), respectively, from the adult head extracts by using the GST tagged kinesin2 tail fragments, KLP64D-T and KLP68D-T.

a non-specific transmembrane protein, was also found to have a distribution profile similar to that of Endo-Orco:GFP (S7C Fig), suggesting that transmembrane receptor proteins are likely to be enriched in the distal OS domain of the olfactory cilia.

At 6 hours AE, the Endo-Orco:GFP domain partly overlapped with the kinesin-2 subunits KLP64D:GFP (Fig 7E and 7F) and KLP68D:YFP (Fig 7G and 7H). Surprisingly, both the KLP64D:GFP (S7D Fig) and the KLP68D:YFP (Fig 7H) peaks were shifted towards the tip of the ciliary OS. Analysis of the full-width-half-maxima (FWHM) separation among the Endo-Orco:GFP, Klp64D:GFP and Jupiter:GFP profiles indicated that the Orco band partly coincides with that of the kinesin-2 subunit KLP64D:GFP at different stages during 0–6 hours AE (Fig 7E and 7F). Also, separation between peaks of their distribution revealed that the Endo-Orco:GFP accumulates much closer to the KLP64D:GFP as compared to the Jupiter:GFP. Together with the FRAP data described above, these observations may suggest that a fraction of ciliary kinesin-2 is engaged in delivering Orco to its parking zone at the distal OS. The anterograde motors are known to inactivate after reaching the distal ends of the cilia and flagella, which may account for the increased accumulation of the kinesin-2 motor subunits at the distal tips. The tip accumulation of KLP64D:GFP and KLP68D:YFP helped to clearly distinguish the tip compartment as separate from that of the Orco domain.

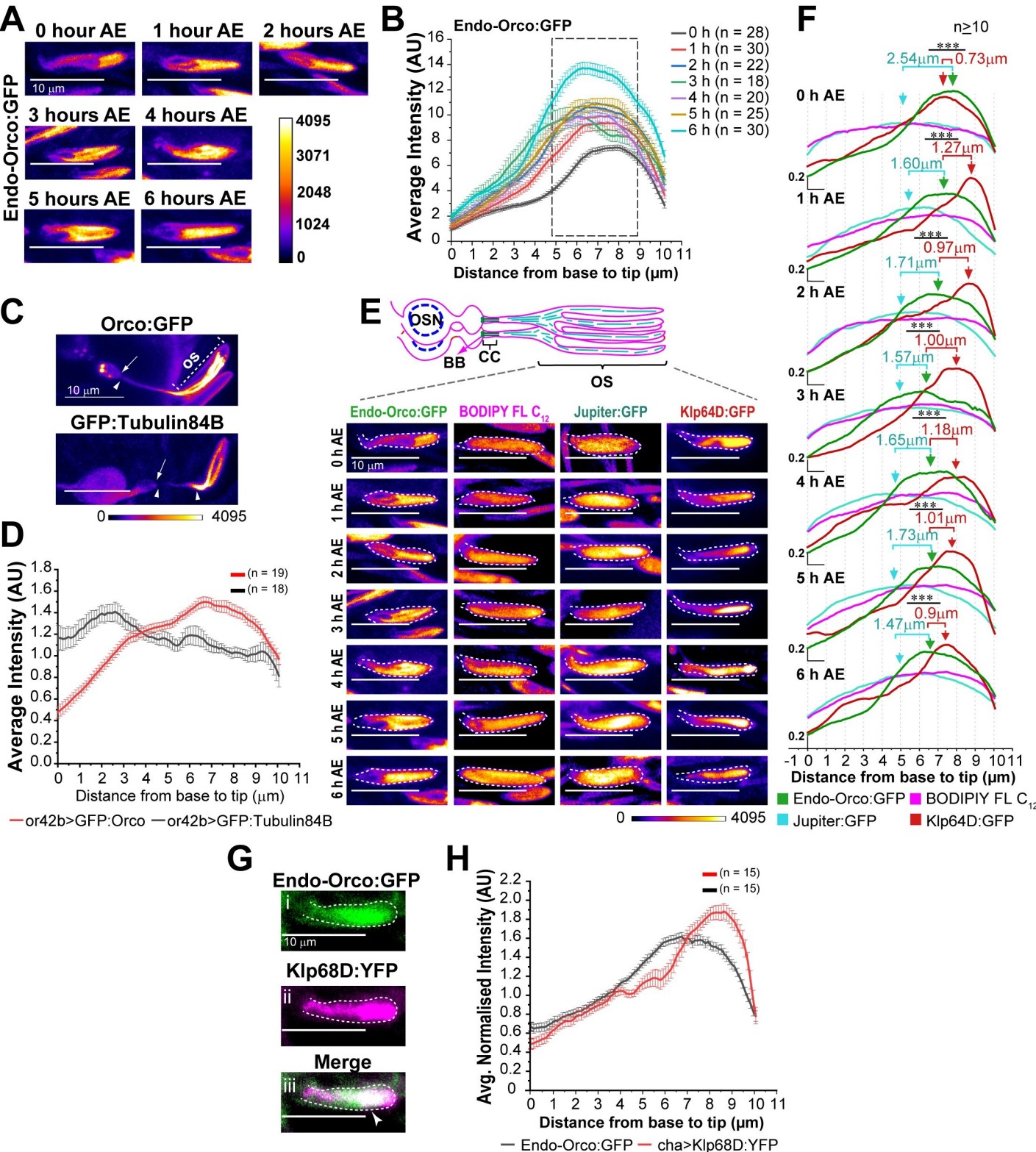

**Fig 7. Orco enrichment domain in the cilia.** (A-B) Fluorescence micrographs (A) and plot profiles (B) of Endo-Orco:GFP distribution (mean ± S.E.M.) along the ciliary OS inside ab1-type *s. basiconica* during 0–6 hours AE. (C, D) GFP:Orco and GFP:Tubulin84B localisation in single cilium expressed on OR42b neuron inside ab1-type sensilla (C) and plot profile of corresponding intensities along the cilia (D). (E, F) Fluorescence micrographs (E) and relative distribution profiles (F) of Endo-Orco:GFP, KLP64D:GFP, Jupiter:GFP and BODIPY FL C₁₂ along the ciliary OS inside ab1-type *s. basiconica* during 0–6 hours AE. The broken line marks cuticle

contour. The green, red, and cerulean arrows indicate peaks of Endo-Orco:GFP, KLP64D:GFP and Jupiter:GFP profiles, respectively, along the cilia. The separations between the peak Endo-Orco:GFP with KLP64D:GFP and Endo-Orco:GFP with Jupiter:GFP are indicated by red and cerulean numerals, respectively. The pairwise significance of difference was estimated using two-tailed Student's T-test, and p-values ($^*$p < 0.05, $^{**}$p < 0.01, and $^{***}$p<0.001) are shown on the plots. (G) Endo-Orco: GFP, KLP68D:YFP colocalisation in the ciliary OS. The GFP and YFP images were obtained by the 445 nm (for GFP) and 515 nm (for YFP) excitations, respectively (see Materials and Methods for details). Scale bars indicate 10 μm. (H) Plot profiles of Endo-Orco:GFP and KLP68D:YFP distribution (mean ± S.E.M.) along the ciliary OS inside ab1-type *s. basiconica*. All images are shown in a false colour intensity heat map (FIRE, ImageJ). Scale for all images—10 μm. Number of sensillae (n) quantified is indicated on the graphs.

## Kinesin-2 is required for packing Orco in the distal domain

We also considered an additional possibility that the interaction with the kinesin-2 motor could actively maintain Orco at the distal part of the ciliary OS. For this purpose, we estimated the total as well as relative intensity distribution of Endo-Orco:GFP along the cilia in control, *Klp64D$^{kj353}$* mutant and kinesin-2 RNAi backgrounds during 0–6 hours AE. The surface plots show that the levels of Endo-Orco:GFP is reduced in the homozygous *Klp64D$^{kj353}$* mutant background (Fig 8A). In the kinesin-2 RNAi backgrounds, the Endo-Orco:GFP fluorescence was highly depleted from the OS domain, particularly at the 3 hours and 6 hours AE, intervening the active transport spurts at 1–2 hours and 4–5 hours AE, respectively (Fig 8A). Also, the FWHM analysis of the length-normalised profile of Endo-Orco:GFP distribution suggested a proximal spreading of the domain (Fig 8B). We also noticed that initially, during 0–3 hours AE, the Endo-Orco:GFP distribution was proximally shifted by a significant margin in the *Klp64D$^{kj353}$* background (S8Ba-d Fig), which was prominent in the kinesin-2 RNAi backgrounds at 6 and 12 hours AE (Figs 8C-b and S8-g), suggesting a comparatively higher penetrance of the Klp64D and Klp68D RNAi as the time progressed.

To further investigate whether the kinesin-2 activity is essential for compacting the Orco domain, we assessed the Endo-Orco:GFP distribution in the heterozygous *Klp64D$^{k5}$* mutant background. *Klp64D$^{k5}$* is an amorphic allele, which carries a missense mutation (Glu551-Lys) at the C-terminal stalk domain of KLP64D. The ciliary OS contained fewer branches bearing singlet microtubules in the *Klp64D$^{k5}$* homozygous background [20]. The EAG responses were, however, not significantly reduced in the *Klp64D$^{k5/+}$* heterozygous background [20]. The KLP64D(E551K) mutation restricts the stalk dynamics [38] and zips the N-terminal stalk in the presence of KAP [39], and thereby, it is expected to disrupt the motor function. Accordingly, one could designate *Klp64D$^{k5}$* as a dominant-negative allele. In *Klp64D$^{k5/+}$* background, the Endo-Orco:GFP distribution spread further towards the proximal domain as compared to the control, *Klp64D$^{kj353}$* and kinesin-2 dsRNA backgrounds (Fig 8C-d), indicating that the compaction of the Orco domain is highly sensitive to the kinesin-2 dose and function. Together, these observations further suggested that interaction between Orco/ ORx and kinesin-2 could maintain them in the designated domain at the ciliary OS.

## Discussion

### Kinesin-2 maintains Orco in a distinct domain at the ciliary outer-segment

The Orco/ORx localisation into the olfactory cilia is critical for the sensory reception [24,40]. Here, we show that the Orco is specifically enriched in the part of the branched OS that is housed inside the porous cuticle shaft. Nano-pores of the cuticle shaft, which houses the ciliary OS containing the branched extensions [20,25], is essential for odour sensing [41]. The branching also increases the surface area of the cilia and accommodates a larger number of the receptors. In addition, packing a large number of branches, typically between 100–140, inside an LB-type shaft would increase the receptor density and facilitate the trapping of cognate odorant molecules at low concentrations. Therefore, the branching of the outer segment and Orco localisation along the branches are crucial for optimising the olfactory responses. We found that the heterotrimeric kinesin-2 (kinesin-2α/β & KAP) motor associates with Orco/

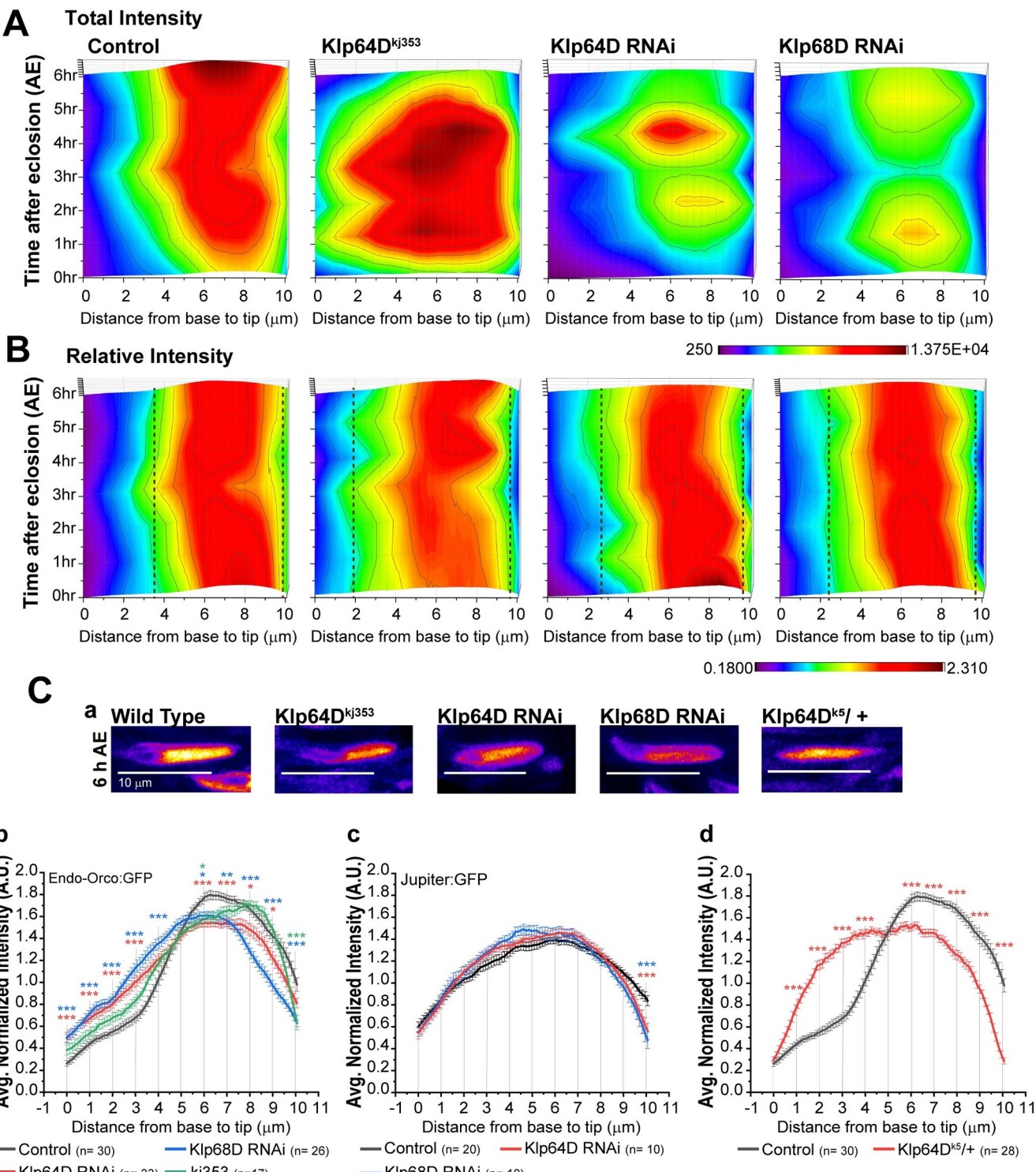

**Fig 8. Endo-Orco:GFP distribution in the kinesin-2 mutant and RNAi backgrounds.** (A-B) Surface plots for total (A) and normalised intensity (B) distributions of Endo-Orco:GFP along the ciliary OS during 0–6 hours AE in the wild-type control, homozygous *Klp64D^kj353* mutant, as well as Klp64D and Klp68D RNAi backgrounds. Dotted lines demarcate the full-width-at-half-maxima (FWHMs) of the Endo-Orco:GFP localisation domains. Note that FWHMs in the mutant backgrounds expands beyond the wild-type domain boundary (4 μm position). (C) Images of Endo-Orco:GFP location in the cilia inside ab1-type sensilla from different genetic backgrounds are shown in the false colour heat map (FIRE, ImageJ) with scale bars indicating 10 μm (a). Plot profiles indicates the Endo-Orco:GFP (b and d) and Jupiter:GFP (c) distribution (mean ± S.E.M.) along the ciliary OS at 6 hours AE in different genetic backgrounds. The pairwise significance of difference was estimated using one-way ANOVA test, p-values (*p < 0.05, **p < 0.01, and ***p<0.001) are shown on the plots. Number of sensillae (n) quantified is indicated on the panels.

ORx complex, helps transfer the protein through the transition zone, and plays a crucial role in localising Orco to this ciliary subdomain in the adult stage (Fig 9). The cytoskeleton of olfactory cilia develops in the pupal stages and remains stable during the early hours after eclosion requiring limited support from the IFT and kinesin-2. Thus, the adult-specific knockdown of kinesin-2 helped identify an independent role of the motor in the Orco transport.

The Orco compartment appeared to be distinct from that of the receptor signalling domains organised at the tips of the primary cilia [14] and that of the cilia on ASER neuron of *C. elegans* [34]. In both these cases, IFT is required for receptor movement. In comparison, the genetic analysis indicated that kinesin-2 might independently control the anterograde transport of Orco and maintain them in an approximately 4 μm domain in the distal OS. The orco domain is separated from that of the kinesin-2 enrichment at the tips with approximately 1 μm offset. The colocalisation study only indicated a partial overlap between the Orco and the tip domain marked by kinesin-2. Therefore, the motor is likely to disengage from Orco once it reaches the designated parking slot in the ciliary membrane.

Further, the FRAP analysis in the kinesin-2 mutant background suggested that Orco diffusion is highly restricted in the cilia. The lipid composition in the ciliary membrane can alter the diffusion coefficient of the transmembrane receptor, such as the Patched-1 [42]. Therefore, altered lipid composition at the distal OS could impede the Orco movement and disengage it from the motor.

The heterotrimeric kinesin-2 is implicated in transporting Rhodopsin [30,43] and Transient Receptor Potential Vanilloid (TRPV) channels [44], via the IFT, although a recent study has contested the claims and suggested that kinesin-2 and IFT function is majorly restricted to the ciliary axoneme assembly [45]. Also, the behaviour analysis in homozygous Ceklp11 and Cekap1 mutant *C. elegans* suggested that the heterotrimeric kinesin-2 could maintain the chemosensory functioning of the AWB and AWC cilia [6,19] through IFT and BBS/Rab8 transports [7]. Cilia maintenance is also IFT dependent [29]. Hence, it has been difficult to distinguish the independent roles of kinesin-2 and IFT in receptor transport. Consistent with this inference, substantial sensory defects were associated with significant cilia growth defects in the recessive, lethal *Klp64D* mutants [20]. The isolation of homozygous viable *Klp64D* mutants with selective olfaction defects and mostly intact cilia further helped identify an independent role of kinesin-2 in the Orco/ORx transport *Drosophila*.

## Role of kinesin-2 in Orco movement across the transition zone

How does the same kinesin-2 complex independently engage in both IFT-based maintenance and Orco/ORx transport? We show that it happens through a time-gated process. The basiconic cilia fully develop during the pupal stages [20]. A previous study showed that compared to the kinesin-2 motor, the ciliary tubulin is highly stable with very low turnover in the adult stage [33], which is confirmed by the low GFP:Tubulin84B enrichment in the cilia when expressed in the adult OSNs using the Orco promoter. Whereas the bulk of the Orco is transported into the cilia at the adult stage immediately after eclosion. Hence, the adult-specific RNAi of kinesin-2 in OSNs had a selectively identifiable impact on Orco transport. The Orco transport in the cilia appears to take place independent of the IFT. The following observations support this assumption. First, an inspection of a large number of images obtained from the *orco^{fTRG}*, *chaGal4>GFP:Orco* and *or42bGal4>GFP:Orco* backgrounds during the 0–6 hours AE failed to identify instances of punctate Orco localisation in the ciliary IS as observed easily in the *chaGal4>Oseg2:GFP* antennae. Also, the loss of Oseg2/IFT172 at the adult stage failed to disrupt the Orco localisation. However, more extensive experiments would be needed to entirely exclude the role of IFT in the Orco transport process.

Further, the Orco/ORx entry is specific to the cilia type, and it is tightly regulated during development. What might be the mechanism of such selectivity? Recent studies suggested that

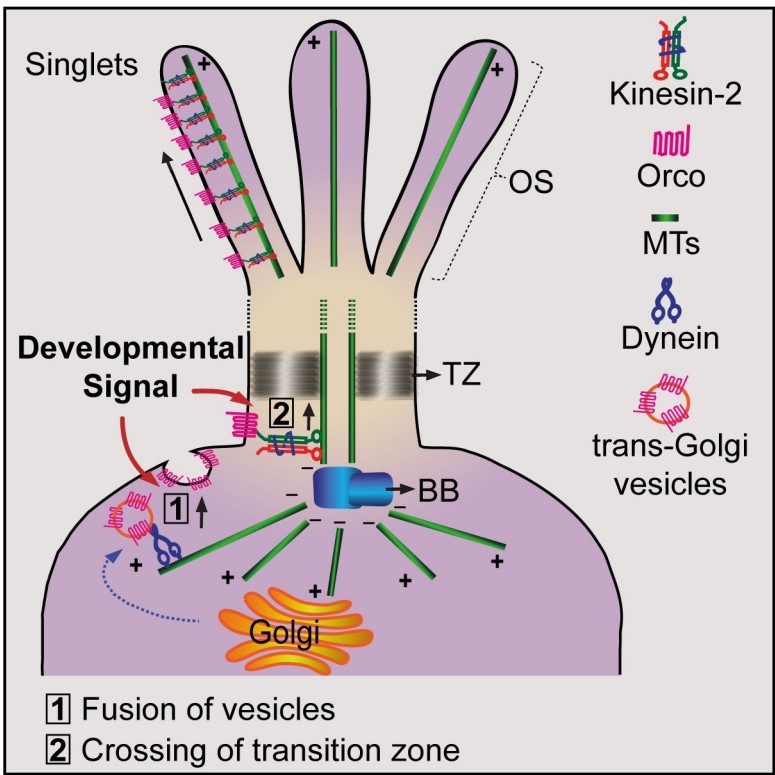

**Fig 9. Model for Orco transport by kinesin-2 motor protein.** Targeting of vesicles carrying receptor proteins from the trans-Golgi network to the ciliary base is brought about by dynein motor and other accessory ciliary trafficking proteins like Rab-GTPases (Rab11, Rab8) along with their GEFS, and also BBSome protein complex. Our results indicate that once the receptor proteins (Orco) are targeted to the ciliary base, heterotrimeric kinesin-2 interacts with Orco and facilitates the transition of the Orco across the ciliary barrier/transition zone into the ciliary OS. Kinesin-2 also helps localise Orco in a defined domain in the distal region of the ciliary OS.

post-translational modifications such as SUMOylation trigger the receptor entry into the cilia [46]. Also, concerted activation of tubby-like protein, exocyst complex and small GTPases like Rab8 regulates transmembrane protein delivery to the ciliary pocket. The Bardet-Biedel Syndrome (BBSome) proteins and the Rab-family of small GTPases plays a critical role in the second stage of this transport [13,47]. The entry and exit through the transition zone are also regulated by cell signalling [48,49]. Therefore, a combination of these functions could deliver Orco/ORx at the base of the cilia. The ciliary membrane is isolated from that of the plasma membrane through an elaborate structure, called ciliary sheet and ciliary necklace [10], at the base of the cilia. Therefore, after the Orco/ORx carrying vesicles fuse at the ciliary pocket, the receptor complex must be actively taken across the barrier. We showed that interaction with the heterotrimeric kinesin-2 motor would be responsible for this movement. It will now be interesting to probe how Orco engages kinesin-2 at the ciliary base.

## Materials and methods

### *Drosophila* culture and stocks

All the fly stocks used in this study are listed in S3 Table. Flies were reared on standard cornmeal agar medium at 25˚C for all the studies. All the experiments were done on dissected antennae of adult flies of different ages as per the experimental requirements.

## Jump assay

Jump tests were performed using a previously established technique [50] with additional modifications described later [51]. Briefly, 2–4 days old, adult flies were anaesthetised on ice and then collected in batches of five in a glass testing tube (length, 14 cm and diameter, 2 cm). The jump tubes were kept vertically with a cotton plug at the bottom, and the tapered end was connected to a chamber containing concentrated odour. As flies walk up the tube wall due to negative geotaxis, a brief (5 seconds) air puff was bubbled through the odour solution and passed into the jump tube using an aquarium pump. The odour puff causes a startling jump response, and the number of flies jumped within 5 seconds of the stimulus was counted as one response. An average of four such jump responses consisting of 20 flies was calculated as one experimental data point, and this was expressed as the percentage jump response. Five of the six kj lines (353, 925, 1072, 1070, and 429) reported significantly reduced jump response by this assay (Fig 2B). All odorants were diluted 1:2 in paraffin oil.

## Ethylmethanesulphonate (EMS) mutagenesis screen (Fig 2A)

Approximately 200 two-day-old isogenised eye colour mutant males ($w^{1118}$) were desiccated for 6–8 hours and then left overnight (~16 hours) on tissue paper soaked with 1% sucrose solution containing 26 mM ethyl methanesulphonate (EMS, Sigma Chemical Co., MO, USA). Subsequently, the EMS-fed males were resuscitated on fresh food for one day and crossed to *w/w*, *klp64D$^{k5}$ h/TM3 Sb e* virgins in 1:4 ratios. The resultant F1 male offspring of the genotype *w; \*/+; \*/Klp64D$^{k5}$ h* (*\*- indicates mutagenised chromosome*) were subjected to jump test with n-butanol. Those of which failed to jump were collected and individually crossed to four *w/w*, *Klp64D$^{k5}$ h/TM3 Sb e* virgins in separate vials. A total of 8000 such lines were set in the F1. The resultant F2 males of the genotype *w; \*/Klp64D$^{k5}$ h* obtained from each line were further subjected to jump tests for enrichment. Only 1400 lines had greater than 30% jump defects. The *w; \*/Klp64D$^{k5}$ h* males from these lines were further outcrossed to *w/w, Klp64D$^{k5}$ h/TM3 Sb e* virgins. The process was repeated for six successive generations to remove the mutated first and second chromosomes. In the end, it yielded seven enriched jump defect lines (Fig 2A). All lines were labelled as kj, post-fixed with an isolation identity number, homozygosed through sibling crosses, and tested for jump response defects to various odours. Of these, five were homozygous viable and consistently failed to jump in response to a multiple different odours (Fig 2B).

## *Drosophila* sample preparation

For live imaging of adult *Drosophila* heads were dissected and mounted in a drop of grade 700 Halocarbon oil (Sigma Chemical Co., MO, USA). For imaging of aged pupal antennae, late third instar larvae were monitored every 30 minutes until they became stationary. It was considered the beginning of the white pupae stage, marked as 0 hours APF, and maintained at 25˚C afterwards. The pupal case was opened, and the head was dissected in Phosphate Buffered Saline (PBS: 137 mM NaCl, 2.7 mM KCl, 10 mM Na2HPO4, 1.76 mM KH2PO4 and pH 7.4) at designated times after the 0 hours APF. The dissected heads were fixed in 4% formaldehyde solution (4% Paraformaldehyde in PBS, pH 7.4) for 30 minutes at room temperature, followed by five washes with PBS. The antennae bearing the second and third segments were then pinched off from the head and mounted in a drop of Vectashield (Vector Laboratories Inc., USA) on a glass slide under a 0.17 mm coverslip.

## Immunostaining

For Orco immunostaining, 1-day old adult *Drosophila* heads were dissected, sectioned in a cryo-microtome (Leica, GmBh) and fixed following standardised protocols. 10 μm sections

were laid on a poly-D-Lysine coated slide and fixed in 4% formaldehyde solution for 30 minutes at room temperature. Then, they were stained using 1:50 dilutions of MAb-Orco (Abcam, USA) and Alexa 647 conjugated goat anti-mouse (Molecular Probes Inc., USA) following the published method [52].

### Dyes microinjection

Both Alexa 546-Dextran (Invitrogen, USA) and BODIPY FL $C_{12}$ (Invitrogen, USA) dyes were injected into the head capsule of anesthetised adults and incubated for 10 minutes to allow the dyes to permeate the third antennal segment through hemolymph. The exclusion of Alexa 546-Dextran from the sensilla confirmed that the ciliary lymph is isolated from the hemolymph (S9B Fig) and the BODIPY label in the sensillum shaft most likely marks the ciliary membrane.

### Image acquisition and analysis

All fluorescence images were obtained under constant acquisition conditions in Olympus (Olympus Imaging Corp., Japan) confocal microscope FV1000SPD using a 63x oil 1.4 NA objective lens or FV3000 using a 60x oil 1.4 NA objective lens. Subsequently, all images were processed using ImageJ (rsweb.nih.gov/1 IJ). All images presented using the false-colour heat map across different time points or genetic backgrounds as placed within a figure panel were collected using identical laser power and gain setting to maintain comparability. Detailed analysis for volume and intensity measurements on 3-D volume-rendered image stacks was done using Imaris 6.1.5 graphics package (Bitplane Scientific Solutions AG, Switzerland). All Figures containing images and line arts were composed in Illustrator (Adobe Inc., USA). The cilia of the OSNs innervating a single sensillum are projected together into the sensory shafts. They are tightly intertwined and could not be resolved using confocal microscopy. Therefore, all the length and volume measurements indicate the combined value of the cilia inside a shaft.

**EAG recording and Electron Microscopy.** described in [20].

### FRAP assay

Adult *Drosophila* heads, severed at the neck region inside a drop of grade 700 halocarbon oil (RI 1.5), were mounted on a coverslip bottom Petri dish with the antennae pressed against the cover glass. The antennae were imaged using 60x Oil 1.4 NA objective and at 3.5x digital zoom using Olympus FV3000 confocal microscope. Multiple, Z-slices (0.25 µm each) were collected to cover the entire ciliary outer segment at 5-minute intervals with 2 µ-seconds pixel-dwelltime. A rectangular ROI (2 µm x 5 µm) at the distal part of the ciliary OS was photobleached for 2 seconds using 400 µW laser power, and the post bleach images of the entire frame were collected at 5-minute intervals for up to 60 minutes. The frame rate was optimised to avoid excessive photobleaching during the post-bleach imaging.

### Estimation of Endo-Orco:GFP and KLP68D:YFP colocalisation

Antennae from Endo-Orco:GFP and *chaGal4>KLP68D:YFP* adults were separately excited in sequence using 445 nm and 514 nm, respectively, and the emissions were collected in 20 nm bands encompassing 495 to 615 nm. KLP68D:YFP emission was negligible in 495–515 nm band when excited using 445 nm. The Endo-Orco:GFP emission was negligible in the 545–565 nm band when excited by 514nm. Therefore, antennae from Endo-Orco:GFP/*chaGal4> KLP68D:YFP* stock was sequentially excited at 445 nm and 514 nm, respectively, and corresponding images were collected in sequence at 495–515 nm and 545–565 nm bands,

respectively, for the colocalisation study. We used Olympus FV3000 confocal microscope and 60x 1.4 NA Oil objective with 3.5x digital zoom.

### *Drosophila* head extracts

Fly heads were manually separated by razor blades, and homogenised in 250 μl of ice-cold, high KCl buffer [19 mM KH2PO4, 81 mM K2HPO4, 400 mM KCl and 1 mM DTT (pH 7.4)], containing the protease inhibitor cocktail (Roche GmbH). Motorised plastic homogeniser and specially designed microfuge tubes were used for this purpose (Pellet Pestle, Kontes, USA). The homogenate was first centrifuged at 17,000 x g for 30 minutes in a refrigerated micro-centrifuge (Biofuge Fresco, Heraeus AG, Germany) to clear the tissue debris and then the supernatant was further centrifuged at 100,000 x g for 1 hour at 4˚C to separate the soluble (supernatant) and membrane-associated proteins (pellet). The pellet was homogenised in 250 μl high KCl buffer containing 1% Triton-X-100 and 1% Sarkosyl, then centrifuged at 100,000 x g for 1 hour at 4 C to extract the membrane-associated proteins, and the resultant supernatant was used as the detergent-extracted fraction for subsequent pulldown experiments.

### Affinity copurification

100 μl aliquots of the soluble and the detergent extracted fractions of the *Drosophila* head homogenates were separately mixed with fixed amounts of purified recombinant proteins in the high KCl buffer. The mixtures containing GST-tagged recombinant proteins were incubated with an equal quantity of Glutathione Sepharose beads for 2 hours at 4˚C. Following this, the Glutathione SepharoseTM beads were washed with several changes of the ice-cold, high KCl buffer containing 0.1 mM Glutathione. Finally, the bound proteins were eluted by incubating the beads in 100 μl proportions of the high KCl buffer containing 1 mM Glutathione. 0.3 ml of the eluted fractions was further incubated with 0.3 ml Glutathione SepharoseTM beads and washed and eluted with equal bed volumes of TMN-D containing 1 mM Glutathione. The buffer conditions and salt concentrations were maintained constant throughout this process. The eluates were boiled in sample loading buffer for SDS-PAGE and western analysis.

### Western blots and immunostaining

The protein mixtures were separated on SDS-PAGE and transferred to PVDF membrane (Hybond-P, Amersham Biosciences Plc. UK) following the supplier's protocol and incubated in different primary antisera solutions (1:1000 dilutions) in 20 mM Tris-buffered saline (TBS, pH 7.4) containing 0.1% TweenR 20. Subsequently, they were incubated either in Rabbit anti-mouse:HRP (dilution, 1:20000; Sigma Chemicals Co. MO, USA), or Goat anti-rabbit:HRP (dilution, 1:20000; Sigma Chemicals Co, MO, USA), in the TBS-T, and developed by using ECLR chemiluminescence detection kit (GE Healthcare Ltd. USA).

### Statistical analysis

All statistical comparisons were carried out using either student's T-test or one way ANOVA with p values calculated according to Bonferroni Test in Origin. P values for data sets in each figure are indicated in the respective figure legends.

### Supporting information

**S1 Fig. Developmental profile of Orco/OR localisation in the cilia on OSNs innervating s. basiconica.** (A) Immunostaining with anti-Orco antibody shows Orco localisation in cilia

inside *s. basiconica* of pupae during 72–96 hours APF and 0–3 hours AE. (B) Endo-Orco:GFP localisation in pupal cilia in s. basiconica from 90–96 hours APF. Arrowheads indicate cell bodies of the OSNs, and arrows mark the ciliary OS. (C) Percentage of sensillae with Endo-Orco:GFP localisation in the ciliary OS of pupal cilia inside *s. basiconica* during 90–96 hours APF. (D) GFP, mCD8:GFP, GFP:Orco and GFP:Or47b localisations in cilia inside *s. basiconica* during 84–96 hours APF; and coexpression of GFP:Orco and GFP:Or43a during 86–96 hours APF. All transgenes were expressed using *chaGal4*. (E) *orcoGal4* expression profile in the antennae during pupal (90–96 hours APF) and adult stages (0–3 hours AE). *UAS-GFP:Orco* was expressed in the OSNs using *OrcoGal4* and the cell body and the cilia (indicated by arrows) were imaged at 90 hour APF, 96 hour APF and 0 day AE. GFP:Orco fluorescence becomes visible in the cell body at 90 hour APF and in the cilia from 96 hour APF. (F, G) Endo-Orco:GFP localisation (F) and total fluorescence intensity (mean ± S.D.) of Endo-Orco: GFP (G) in the ciliary OS inside *s. basiconica* of adult flies during 0–144 h AE. The pairwise significance of difference was estimated using one-way ANOVA test, p-values ($^*$p < 0.05, $^{**}$p < 0.01, and $^{***}$p<0.001) are indicated on the plots. Error bars represent as ± SD. All images are shown in the false colour intensity heat map (FIRE, ImageJ). Scale for images (A), (B) and (C) 10 μm; (E) 5 μm. The number of sensillae (n) quantified are indicated on the graphs. Note: Images shown in panel F and quantification shown in panel G were collected using Olympus FV1200 confocal microscope using different laser power, HV and gain settings as compared to the images of Endo-Orco:GFP shown in other figures, which were collected on FV3000 using uniform laser power and HV and gain settings. Therefore, the intensity values can only be compared amongst the histograms of the panel (F) only.
(TIF)

**S2 Fig. Volume of the cilia on Or42B OSN and *or42B*-Gal4 expression profiles in the ab1-type *s. basiconica*.** (A) Adult antenna expressing Endo-Orco:GFP and *or42b>tdTomato* (*or42bGal4/UAS-tdTomato*) marking the ab1-type sensilla (arrowhead). (B, C) Ciliary volume (mean ± S.D.) marked by tdTomato expressed in all OSNs (B) and the tdTomato fluorescence intensity (mean ± S.D.) in the cilia of Or42b specific OSNs (C) during 0–6 hours AE. The pairwise significance of difference was estimated using one-way ANOVA test, p-values ($^*$p < 0.05, $^{**}$p < 0.01, and $^{***}$p<0.001) are shown on the plots. Images are shown in a false colour intensity heat map (FIRE, ImageJ). Scale bars indicate 50 μm (A) and 2 μm (B, C), respectively. The number of sensillae (n) quantified are indicated on the graphs.
(TIF)

**S3 Fig. Mapping of *kj* alleles by genetic non-complementation and rescue analysis.** (A-B) The chordotonal cilia are unaffected in homozygous Klp64D$^{kj}$ backgrounds. TEM images of cross-sections of scolopidia from Johnston's Organs in wild-type and homozygous mutant antennae (A). Each scolopidium contained two sensory cilia (arrows), with 9+0 organisation of microtubule (inset). The chordotonal cilia, appeared normal in the mutants, play essential roles in proprioception and negative geotaxis. (B) Homozygous *Klp64D$^{kj}$* alleles are coordinated. Negative geotaxis (test of coordination) of wild-type and homozygous *kj* adults was measured by estimating the relative number of flies above the half-length of a 30 cm cylinder 5 minutes after banging. Homozygous *w1118* was used as the wild-type control. It showed that they were coordinated like the wild-type flies. (C-E) EAG response defects of the homozygous *Klp64D$^{kj}$* mutants are not rescued by the OSN-specific expression of the *KLP68D* transgene. Histograms indicate mean (± S.E.M) electroantennogram (EAG) responses from the antennae of various mutant combinations. *Klp64D$^{k1}$* is a lethal total-loss-function allele of *Klp64D* and *Klp68D$^{P1}$* is a hypomorphic, viable, P-element insertion allele of *Klp68D*. *chaGal4* induced expression of *UAS-KLP68D:YFP*, as well as *UAS-KLP64D:GFP* in OSNs rescued the cilia

development and EAG defects of the homozygous *Klp68D^{P1}* (Jana et al., 2011). The data shown in panel E indicate that loss of KLP68D could partly compensate by the overexpression of KLP64D. The number of antennae/flies (N) quantified are indicated on the graphs.
(TIF)

**S4 Fig. Ciliary volume and GFP:tubulin localisation in *Klp64D^{kj353}* homozygous mutant backgrounds.** (A) Total fluorescence intensity (a) and volume (mean ± S.E.M.) of ciliary OS (b) inside *s. basiconica* marked by the soluble eGFP, membrane-associated mCD8:GFP, the cytoskeleton marker GFP:Tubulin84B and GFP:Orco (all driven by *chaGal4*) at two-day AE in control and homozygous *Klp64D^{kj353}* backgrounds. The pairwise significance of difference was estimated using a two-tailed Student's T-test, and the p-values (*p < 0.05, **p < 0.01, and ***p<0.001) are indicated on the bars. (B) GFP:Tubulin84B localisation in the ciliary OS of ab1-type *s. bascionica* in the *chaGal4>UAS-GFP:Tubulin84B, or42bGal4>UAS-GFP:Tubulin84B*, and *orcoGal4>UAS-GFP:Tubulin84B*. Note that very little GFP:Tubulin84B is transported into the cilia when expressed in the adult stage by *orcoGal4*. The expression of *chaGal4* starts at 30 hours APF much before the cilia growth, and that of or42bGal4 is found to begin at 60 hours APF. The adult fly emerges at 105 hours APF at 25˚C. All images are shown in the false colour intensity heat map (FIRE, ImageJ). Scale for images—10 μm. The number of sensillae (n) quantified are indicated on the graphs.
(TIF)

**S5 Fig. Endo-Orco:GFP enrichment in the ciliary OS in kinesin-2 RNAi and mutant backgrounds.** (A) Endo-Orco:GFP localisation inside the ab1-type *s. basiconica* from Control (*Endo-Orco:GFP; +*), and homozygous *Klp64D^{kj353}* mutant (*Endo-Orco:GFP; Klp64D^{kj353}*) backgrounds at 6 days AE (a). Histograms depict total fluorescence intensity (mean ± S.D.) of Endo-Orco:GFP in Control, and *Klp64D^{kj353}* mutant backgrounds at 6 days AE (b). The pairwise significance of difference was estimated using the two-tailed Student's T-test, and p-values (*p < 0.05, **p < 0.01, and ***p<0.001) are indicated on the plots. Error bars represent ± S.D. (B) Ciliary morphology is affected due to the prolonged absence of the kinesin-2 subunits in the RNAi backgrounds. Fluorescence micrographs depicting the distribution of Endo-Orco:GFP in the ciliary OS of the Control, Klp64D RNAi and Klp68D RNAi at 1 day and 2 days AE. Since the ciliary structure seemed disrupted (presence of hollow pocket-arrows) at 1 day and 2 days AE, we did not estimate Endo-Orco:GFP levels in the ciliary OS beyond 12 hours AE. (C) Kinesin-2 knockdown does not affect the cytoskeleton of the cilia inside *s. basiconica*. Jupiter:GFP localisation in the ab1-type s.basiconica from control (*orcobGal4/+; Jupiter:GFP/UAS-Dicer*), Klp64D RNAi (*orcoGal4/UAS-Klp64D RNAi; Jupiter:GFP/UAS-Dicer*) and Klp68D RNAi (*orcoGal4/UAS-Klp68D RNAi; Jupiter:GFP/UAS-Dicer*) background at 6 hours AE (a). Histograms depict total Jupiter:GFP fluorescence intensity (mean ± S.D.) in Klp64D and Klp68D RNAi backgrounds at 6 hours AE (b). The pairwise significance of difference was estimated using the one-way ANOVA test, p-values (*p < 0.05, **p < 0.01, and ***p<0.001) are indicated on the plots. All images are shown in the false colour intensity heat map (FIRE, ImageJ). Scale for images—10 μm (A-a, and C-a); and 2 μm (B). The number of sensillae (n) quantified are indicated on the graphs.
(TIF)

**S6 Fig. Endo-Orco:GFP transport into the olfactory cilia in the adult-specific RNAi of kinesin-2 and in the homozygous *Klp64D^{kj353}* mutant backgrounds.** (A) Fluorescence recovery of Endo-Orco:GFP after photobleaching (FRAP) in the ciliary OS in *s. basiconica* of the control (*orcoGal4, Endo-Orco:GFP/+; UAS-Dicer/+*), Klp64D RNAi (*orcoGal4, Endo-Orco:GFP/UAS-Klp64D RNAi; UAS-Dicer/+*), Klp68D RNAi (*orcoGal4, Endo-Orco:GFP/*

*UAS-Klp68D RNAi; UAS-Dicer/+*) and homozygous *Klp64D^{kj353}* mutant (*Endo-Orco:GFP; Klp64D^{kj353}*) backgrounds during 4–5 hours AE. All images are shown in the false colour intensity heat map (FIRE, ImageJ), and scale indicates 2 μm. (B) Relative FRAP profiles are plotted for individual *s. basiconica* in different genetic backgrounds. Note that the recovery profiles in control were unimodal. It reached saturation after 15 minutes in a majority of cases. However, in a significant number (n = 3 of 11) of cases, a second spurt occurred in the mid-course, which enhanced the recovery beyond 100%, suggesting that Orco is perhaps transported in 15–20 minute spurts during 4–5 hours AE. The loss of kinesin-2 significantly slowed the recovery. The number of sensillae (n) quantified are indicated on the graphs.
(TIF)

**S7 Fig. Comparison of the domain of Orco accumulation with that of cytoskeleton, membrane and kinesin-2 motor.** (A) Ciliary localisation of Endo-Orco:GFP, cytoskeleton markers- GFP:Tubulin84B and Jupiter:GFP; soluble protein- eGFP; membrane marker- mCD8: GFP; and ciliary trafficking machinery- Klp64D:GFP and Oseg2:GFP, at 6 hours AE. Except for Jupiter:GFP (protein trap) and Endo-Orco:GFP (fosmid line), all the oher transgene expressions were driven by the *chaGal4*. (B-D) Plot profiles of the distribution (mean ± SEM) of Endo-Orco:GFP and the other ciliary markers along the ciliary OS inside ab1-type s. basiconica at 6 hours AE. The pairwise significance of difference was estimated at every 1 μm interval along the length of the cilia using one-way ANOVA test, p-values ($^*p < 0.05$, $^{**}p < 0.01$, and $^{***}p < 0.001$) are indicated on the plots. All images are shown in the false colour intensity heat map (FIRE, ImageJ). Scale for images- 10 μm. Number of sensillae (n) quantified are indicated on the graphs.
(TIF)

**S8 Fig. Endo-Orco:GFP enrichment domains in the wild-type and kinesin-2 mutant backgrounds.** (A) Fluorescence intensity micrographs of Endo-Orco:GFP along the ciliary OS of ab1-type *s. basiconica* during 0–5 hours AE due to the knockdown of kinesin-2 motor subunits–KLP64D and KLP68D. and in the homozygous *Klp64D^{kj353}* backgrounds, respectively. All images are shown in the false colour intensity heat map (FIRE, ImageJ). Scale for images- 10 μm. (B) Comparison of the relative distribution of Endo-Orco:GFP (mean ± S.E.M) during 0–5 hours AE and 12 hours AE in control, KLP64D RNAi, KLP68D RNAi, and homozygous Klp64D kj353 mutant backgrounds. The pairwise significance of the differences was estimated at a few sample points along the ciliary OS using a one-way ANOVA test; p-values ($^*p < 0.05$, $^{**}p < 0.01$, and $^{***}p < 0.001$) are indicated on the plots. The number of sensillae (n) quantified are indicated on the graphs.
(TIF)

**S9 Fig. Estimating the length of the ciliary OS.** (A) Fluorescence micrographs of cilia in *s. bascionica* showing localisation of Oseg-2:GFP, GFP:Tubulin84B, Orco:GFP^{fTRG} (Endo-Orco: GFP), DIC of the cuticle of *s. basiconica*, and merge of DIC and Endo-Orco:GFP. The starting point of the ciliary OS (white arrowheads) coincides with the base of the cuticle shaft and are always marked by a characteristic bend. EndoOrco:GFP mostly localises after this bend (white arrowhead), whereas Oseg-2:GFP was found to localise from the basal body (yellow arrowhead) through the inner segment and up to the middle region of the ciliary OS. The distal-most end of the ciliary OS marks the endpoint (white arrows) for measurement of the plot profiles. The length along which the line segment was drawn to measure the plot profiles is represented by a dark grey dotted line. All images are shown in the false colour intensity heat map (FIRE, ImageJ). (B) Localisation of Alexa 546-Dextran is excluded from the OSNs and sensillae. (a) Alexa 546-Dextran marked the coelom of the third antennal segment. (b) Inverted

image of Alexa 546-Dextran indicates its exclusion form the sensillum shaft region. (c) Endo-Orco:GFP marking the OSNs and the *s. basiconica*. (d) Merge of Alexa 546-Dextran and Endo-Orco:GFP depicting that Dextran is excluded from the lymph surrounding the olfactory sensory cilia innervating the sensillae.
(TIF)

**S1 Table. The chemotactic behaviour of homozygous *Klp64D^{kj}* mutant larvae.** Olfactory responses for third instar larvae from wildtype, *Klp64D kj* mutants, and *Or83b/Orco* null (positive control) were measured by performing the larval plate assay. Third instar larvae were placed in the center of petriplate with odour on one half of the plate and diluent on the other half. They were given a choice to move towards either direction. The larvae on each side was calculated and the response index was calculated for each genotype. *Klp64D kj* mutant larvae did not show any olfactory response defects towards the odours when compared to wildtype. While *Or83b/Orco* null mutants showed a marked decrease in response, as expected. *Note*: *n-vale indicates total number of runs counted. Each run lasted for 15 minutes and consisted of approximately 20 larvae.*
(DOCX)

**S2 Table. Allelic complementation analysis of *kj* isolates with known *Klp64D* alleles and the transgenic rescue of the defect, estimated using elactroantennogram (EAG).** EAG values were obtained from 2-days old adult flies of specific genotypes. Values in each cell indicate mean (± S.D.) responses in mV (N≥6) for each genotype. Pairwise significance of differences was calculated with respect to the wild type control values for the respective odours using two tailed Student's T-test. Significant (*, p<0.1) and very significant (**, p<0.01) differences are indicated in the cells. Color key: **red—Ethyl Acetate (EA)**, **blue n-Butanol (n-But)**
(DOCX)

**S3 Table. List of fly stocks used.** The fly stocks, genotypes and source information is listed in this table.
(DOCX)

**S1 Data. The raw data used for the plots depicted in Figs 1–5 and S1–S7.**
(XLSX)

**S2 Data. The raw data used for the plots depicted in Fig 7.**
(XLSX)

**S3 Data. The raw data used for the plots depicted in Fig 8.**
(XLSX)

**S4 Data. The raw data used for the plots depicted in S8 Fig.**
(XLSX)

**S5 Data. Western blots of panels presented in Fig 6.**
(PDF)

## Acknowledgments

We thank Prof. Tomer Avidor-Reiss, University of Toledo, Texas, USA, for the *UAS-Oseg-2*: *GFP* and *UAS-Oseg4*:*GFP* fly stock, Dr. Manish Agarwal (TIFR-H) and Dr. Debasis Das (TIFR) for sharing antibody reagents, and B. Karmakar and R. Phadke for technical help with the instrumentation and experiments. We thank K V Boby and Lalit Borde for their assistance with the confocal and TEM imaging. We especially acknowledge Prof. Monica Bettencourt-

Dias, IGC, Portugal, for exceptionally generous support. A part of the experiments was carried out at IGC by SCJ.

## Author Contributions

**Conceptualization:** Swadhin Chandra Jana, Priya Dutta, Akanksha Jain, Krishanu Ray.

**Data curation:** Swadhin Chandra Jana, Priya Dutta, Akanksha Jain, Anjusha Singh, Lavanya Adusumilli, Mukul Girotra, Seema Shirolikar, Krishanu Ray.

**Formal analysis:** Swadhin Chandra Jana, Priya Dutta, Akanksha Jain, Anjusha Singh, Lavanya Adusumilli, Krishanu Ray.

**Funding acquisition:** Krishanu Ray.

**Investigation:** Swadhin Chandra Jana, Priya Dutta, Akanksha Jain, Anjusha Singh, Lavanya Adusumilli, Mukul Girotra, Diksha Kumari, Seema Shirolikar, Krishanu Ray.

**Methodology:** Swadhin Chandra Jana, Priya Dutta, Mukul Girotra, Diksha Kumari, Seema Shirolikar, Krishanu Ray.

**Project administration:** Krishanu Ray.

**Resources:** Swadhin Chandra Jana, Krishanu Ray.

**Supervision:** Krishanu Ray.

**Validation:** Swadhin Chandra Jana, Priya Dutta, Akanksha Jain.

**Writing – original draft:** Swadhin Chandra Jana, Priya Dutta, Akanksha Jain, Krishanu Ray.

**Writing – review & editing:** Swadhin Chandra Jana, Priya Dutta, Akanksha Jain, Krishanu Ray.

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
