## [Decision Letter · Decision Letter 0]

10 Mar 2021

Dear Dr Ray,

Thank you very much for submitting your Research Article entitled 'Kinesin-2 transports Orco into the olfactory cilium of Drosophila melanogaster at specific developmental stages' to PLOS Genetics.

The manuscript was fully evaluated at the editorial level and by independent peer reviewers. The reviewers appreciated the attention to an important topic but identified some concerns that we ask you address in a revised manuscript.

We therefore ask you to modify the manuscript according to the review recommendations. Your revisions should address the specific points made by each reviewer.

[LINK]

Yours sincerely,

Liliane Schoofs

Associate Editor

PLOS Genetics

Gregory P. Copenhaver

Editor-in-Chief

PLOS Genetics

Reviewer's Responses to Questions

**Comments to the Authors:**

Reviewer #1: This manuscript was an absolute joy to read. The authors show that the heterotrimeric kinesin 2 is responsible for the post-IFT trafficking of the odorant receptors into the distal olfactory cilium, in two specific phases in early adulthood. The experiments are well designed and systematically performed to the conclusions clear and indisputable.

I have only very small comments and a number of typos and grammatical changes that the authors should consider.

Page 2, abstract. Line 20. I suggest to add “The”. Thus, “The cilium, the sensing centre….”

Line 31. Delete “a”. “…the loss of critical components….”

Page 3, Author Summary. Line 39. “organelles” should be plural in agreement with “Cilia”.

Line 42. Delete “the”. “…such as photoreception,…”

Page 4. Line 75. “direction” should be singular.

Page 5. Line 102. By “unprecedented”, do the authors mean “described here for the first time”? Or do they mean that such temporal shifts do not happen earlier in development.

Line 106. “associates” should be “associate”, for verb agreement.

Page 6. Line 112. Insert “types” after “sensilla”.

Page 7. Line 133. “its” should be “their” for agreement with the plural “sensilla”.

Line 147. I expect the authors will want to remove the reference to Jain’s thesis.

Page 8. Line 164. I could not find any description in the methods section for how the jump test was performed for the screen. The figure shows an outline, but the experimental details should be included in the methods sections. For example, what odorant was used? How was it presented? How often? How was the scoring done?

Line 169. I believe the data mentioned here are in Figure S3, not S2.

Page 9. Line 180. Spelling of “coeloconica”.

Lines 182-184. The statement that the EAGs “indicated that the loss of kinesin-2 could suppress odour receptor localization” is over-stated. This should be presented as an inference, rather than as a conclusion. For example, the authors may wish to change “indicated” to “suggested”.

Page 11. Line 223. Stating the question in the negative is a bit awkward, and suggests a preconceived conclusion. I suggest “To test whether endogenous Orco transport by kinesin-2 depends on IFT, …”

Line 225. Delete “the”: “…just before eclosion”.

Line 227. Remove reference to Jain’s thesis? Or include the citation in the bibliography.

Line 236-237. The data are in Figure S5, not S4.

Line 241. Again, S5.

Page 12. Line 258. Verb agreement: “GFP:Orco and OR47b were selectively…”

Page 13. Lines 282-283. The authors point to the similarity of Endo-Orco:GFP and KLP64D:GFP. However, in the figure, KLP64D:GFP does appear to be even more distal than Endo-Orco:GFP. This point does not seem to be addressed by the authors, even in the discussion. Is it possible that this difference could be explained, for example, by the idea that when Orco is released in the distal cilium membrane, it can drift proximally, while the motor protein does not? This would seem to be an important point to address in the discussion.

Page 14. Line 304. “…distribution of in the…” needs to be corrected.

Line 306. The word “microtubules” seems to be missing: “…fewer singlet branched microtubules in the…”

Line 307. “responses” in the plural.

Line 312. “observations” in the plural.

Page 15. Line 325. “helps” for verb agreement.

Page 16. Line 340. Spelling of “Orco/ORx”

Page 18. Lines 386-389. These last 2 sentences of this section do not belong here, as they are results and conclusions. They should be moved to the appropriate sections.

Line 388. Spelling of “BODIPY”

Page 26. Figure 1 legend. Line 630. “show” for verb agreement.

Page 29. Line 665. “driven by orcoGal4 (i) and chaGal4 (ii)”. Is it possible for the authors to switch these 2 subpanels, for parity with part a), where chaGal4 is presented first, then orcoGal4?

Line 666 and line 669 on the next page. Please include the (i) and (ii) indicators for the subpanels. Otherwise it looks strange to have “c)” twice.

Page 34. Bottom of panel f. Spelling of “BODIPY” needs to be corrected.

Page 42. Line 770. Spelling of “scolopidia”.

Line 771. Spelling of “scolopidium”

Line 773. “anti-geotropic walk” seems like unconventional terminology. “negative geotaxis” is more conventional, as used in the next line.

Page 44. Line 787-788. The scale bar reference should be moved to part a) of the legend, as there are no scale bars in panel b).

Page 46. Line 796. Delete “in”: “unaffected upon adult specific”.

Reviewer #2: In this manuscript Jana et al., report their efforts to understand the mechanisms that localize the odor-receptor coreceptor (Orco) into the cilia of the olfactory sensory neurons. The authors find that kinesin-2 transports Orco into the cilium, and this activity is independent from its function in IFT. In addition, they show that Orco enters cilia of the olfactory sensory neurons at a specific developmental time following the eclosion, in two separate bursts. Once in the cilium, kinesin-2 restricts Orco’s localization to a specific ciliary subdomain.

Using genetic approaches, the authors were able to dissect the function of kinesin-2 to study its role in transport of transmembrane receptors into the cilia, independently of IFT and cilia development. Although, this study does not provide much mechanistic insight into how kinesin-2/Orco entry is regulated, the authors show compelling evidence that kinesin-2 plays a complex role in cilia of olfactory neurons. As ciliary signaling and establishment of unique signaling domains within ciliary compartments is still poorly understood, the work by Jana et al., provide valuable insight into the transport and localization of a transmembrane receptor, Orco into the cilia of olfactory neurons in adult flies.

However, some revisions and additional experiments are needed before the manuscript can be accepted for publication.

1. Introduction needs revisions, the authors should provide sufficient context for presented background, for example while explaining Smo transport, the authors state: ‘Whereas the kinesin-family motor Costal2/Kif7, which is immotile and does not bind to either Smo or Gli, is implicated in establishing the signalling-competent […]’, the role of Gli in Smo’s transport/function is unclear.

2. All abbreviations should be explained immediately as they appear.

3. Figure legends, for both main figures and supplementary materials need major revisions; for example, repeated statements in figure S4. In addition, figure legends should follow the same format. Presented data should be adequately explained in figure legends, for example meaning of the arrowheads.

All figures should be properly called in the text, for example, there is no reference for Figure 1F in the text, there are two figures S6, figure S3C, and entire figure S2 is incorrectly referenced.

4. The authors should provide information regarding the number of experimental repeats used to perform statistical analysis and calculate significance.

5. All images, including zoomed panels should have scale bars.

6. The authors have generated 5 mutant strains showing selective disruption of olfaction in adult flies, however, only 2 were used for subsequent studies. The authors should provide rationale for focusing on particular mutants.

7. The authors use intensity heat maps to present data, the authors should specify whether they are uniform across images from different time points.

8. The authors state ‘We reasoned that Orco gene expression from 80-h APF [29] onwards could restrict the entry phase.’ This statement is unclear and should be re-phrased.

9. The authors find that one of the mutants with impaired olfaction used in this study have ‘ […] a small but significant reduction in the number of singlet microtubules […]’ yet the authors state that these results are consistent with lack of effect on tubulin level in the mutant cilia. Therefore, the interpretation of the data is unclear and should be adjusted.

10. The authors find that 2 used in this studies mutants with impaired olfaction show reduced localization of Orco in cilia, however it is not clear what effect selected mutations have on Orco association with kinesin-2. Especially, that based on the presented data it appears that Orco’s level in cilia still increases over time, even in the mutant flies. Therefore, the authors should test whether the generated mutant show decrease in Orco-kinesin- interaction and/or impaired transport.

11. Based on the presented data, it appears that signal of kinesin-2 only partially overlaps with that of Orco in cilia of olfactory neurons after eclosion (figure 6f). Given that presented measurements are based on the measurements of signal intensity over length of cilia in fly lines expressing GFP-tagged proteins of interest; it would be informative if the authors could supplement these observations with true co-localization studies.

12. The authors show that distribution of Orco in kinesin-2 mutant and RNAi flies is not restricted to a ciliary sub-compartment, as in control flies after 6 hours following the eclosion. The authors however do not show whether in these backgrounds localization of Orco to a ciliary domain is truly impaired or simply delayed.

13. The authors should include full western blots presented on figure 5 for evaluation.

**Have all data underlying the figures and results presented in the manuscript been provided?**

Reviewer #1: Yes

Reviewer #2: Yes

PLOS authors have the option to publish the peer review history of their article (what does this mean?). If published, this will include your full peer review and any attached files.

Reviewer #1: No

Reviewer #2: No

---

## [Editor Report · Decision Letter 1]

29 Jul 2021

Dear Dr Ray,

We are pleased to inform you that your manuscript entitled "Kinesin-2 transports Orco into the olfactory cilium of Drosophila melanogaster at specific developmental stages" has been editorially accepted for publication in PLOS Genetics. Congratulations!

Yours sincerely,

Liliane Schoofs

Associate Editor

PLOS Genetics

Gregory P. Copenhaver

Editor-in-Chief

PLOS Genetics

Comments from the reviewers (if applicable):

**Data Deposition**

http://datadryad.org/submit?journalID=pgenetics&manu=PGENETICS-D-21-00072R1

**Press Queries**

---

## [Editor Report · Acceptance letter]

16 Aug 2021

PGENETICS-D-21-00072R1 

Kinesin-2 transports Orco into the olfactory cilium of *Drosophila melanogaster* at specific developmental stages 

Dear Dr Ray, 

We are pleased to inform you that your manuscript entitled "Kinesin-2 transports Orco into the olfactory cilium of *Drosophila melanogaster* at specific developmental stages" has been formally accepted for publication in PLOS Genetics! Your manuscript is now with our production department and you will be notified of the publication date in due course.

With kind regards,

Agnes Pap

PLOS Genetics

On behalf of:
